# Stochastic parametric skeletal dosimetry model for humans: General approach and application to active marrow exposure from bone-seeking beta-particle emitters

**Marina O. Degteva**[1], **Evgenia I. Tolstykh**[1], **Elena A. Shishkina**[1,2], **Pavel A. Sharagin**[1], **Vladimir I. Zalyapin**[3], **Alexandra Yu. Volchkova**[1], **Michael A. Smith**[4], **Bruce A. Napier**[4]*

**1** Urals Research Center for Radiation Medicine, Chelyabinsk, Russia, **2** Chelyabinsk State University, Chelyabinsk, Russia, **3** South Urals State University, Chelyabinsk, Russia, **4** Pacific Northwest National Laboratory, Richland, Washington, United States of America

* Bruce.Napier@pnnl.gov

## Abstract

The objective of this study is to develop a skeleton model for assessing active marrow dose from bone-seeking beta-emitting radionuclides. This article explains the modeling methodology which accounts for individual variability of the macro- and microstructure of bone tissue. Bone sites with active hematopoiesis are assessed by dividing them into small segments described by simple geometric shapes. Spongiosa, which fills the segments, is modeled as an isotropic three-dimensional grid (framework) of rod-like trabeculae that "run through" the bone marrow. Randomized multiple framework deformations are simulated by changing the positions of the grid nodes and the thickness of the rods. Model grid parameters are selected in accordance with the parameters of spongiosa microstructures taken from the published papers. Stochastic modeling of radiation transport in heterogeneous media simulating the distribution of bone tissue and marrow in each of the segments is performed by Monte Carlo methods. Model output for the human femur at different ages is provided as an example. The uncertainty of dosimetric characteristics associated with individual variability of bone structure was evaluated. An advantage of this methodology for the calculation of doses absorbed in the marrow from bone-seeking radionuclides is that it does not require additional studies of autopsy material. The biokinetic model results will be used in the future to calculate individual doses to members of a cohort exposed to [89,90]Sr from liquid radioactive waste discharged to the Techa River by the Mayak Production Association in 1949–1956. Further study of these unique cohorts provides an opportunity to gain more in-depth knowledge about the effects of chronic radiation on the hematopoietic system. In addition, the proposed model can be used to assess the doses to active marrow under any other scenarios of [90]Sr and [89]Sr intake to humans.

**Data Availability Statement:** All relevant data are within the manuscript and its Supporting information files.

**Funding:** This work was funded by Federal Medical-Biological Agency of Russia (MOD, EIT, EAS, PAS, VIZ) and the U.S. Department of Energy's Office of International Health Programs (MAS, BAN) in the framework of joint US-Russia JCCRER Project 1.1 (https://www.energy.gov/ehss/russian-health-studies-program).

**Competing interests:** The authors have declared that no competing interests exist.

## Introduction

Environmental radioactive contamination in the Southern Urals from Mayak plutonium facility releases in the 1950s led to substantial doses to the residents of nearby communities, and, subsequently, to health effects [1–6]. In particular, cases of chronic radiation syndrome (considered to be an early deterministic effect of red bone marrow exposure) were diagnosed in residents using the Techa River, contaminated by the Mayak discharges [7]. An increase of leukemia associated with radiation dose to bone marrow was noted in the cohort of about 30,000 Techa riverside residents [8, 9] and also in the combined cohort of about 20,000 persons exposed *in utero* whose mothers lived by the Techa River or worked at Mayak [10]. Excess relative risk of leukemia was also observed in a cohort of about 20,000 residents exposed as a result of an accident that occurred at Mayak in 1957, but it was not statistically significant [2]. Further study of these unique cohorts provides an opportunity to gain more in-depth knowledge about the effects of chronic radiation on the hematopoietic system.

Evaluation of individual dose and associated uncertainty from ingested $^{90}$Sr and also the shorter-lived $^{89}$Sr is of utmost importance in the dosimetric support of the epidemiological studies [4]. Strontium-90 was the most significant radionuclide for the exposed residents in terms of bone marrow dose. Maximum lifetime intake of this radionuclide was estimated to be 2-3 MBq from Techa River use in the early 1950s [11]. Monitoring of $^{90}$Sr-body burden in Urals residents has continued for over 60 years, and more than 20,000 people have been examined [12–14].

Calcium-like $^{89}$Sr and $^{90}$Sr incorporated in the mineralized matrix of bone are a source of beta radiation for the marrow. An age- and gender-dependent biokinetic model [15] was developed to estimate temporal changes of concentrations of $^{89,\,90}$Sr in trabecular and cortical bones after radionuclide intakes. Until recently, to convert strontium concentration into marrow dose rate, we used so called "path length-based" methods of radiation particle transport described in [16, 17]. Later studies [18, 19] supported the use of voxel-based skeletal models permitting more accurate estimates of energy deposition in marrow targets.

The current paper presents a general description of the voxel-based Stochastic Parametric Skeletal Dosimetry (SPSD) model for humans, originally proposed by E. Shishkina and V. Zalyapin [20, 21]. This SPSD model accounts for the individual variability of the macro- and microstructure of bone tissues and is currently implemented for males and females of different ages beginning from birth.

## Hematopoietic sites of human skeleton as objects for modeling

Active marrow (*AM*) is hematopoietically active and gets its red color from the large numbers of erythrocytes (red blood cells) being produced. AM sites of the skeleton correspond to trabecular spongiosa regions covered by a cortical shell (cortex). The spongiosa contains both source areas (bone trabeculae) and target area (active marrow).

### Distribution of AM through the human skeleton

At birth, the whole skeleton is filled with *AM*. Throughout life there is a conversion of much *AM* to inactive (yellow) marrow *(IM)*. As a result of skeletal growth and AM reductions at different age periods, the skeletal hematopoietic sites contain both red and yellow marrow in different proportions. The location of hematopoietic sites in humans of different ages is shown in Fig 1. The age-related changes in the hematopoietic sites and quantitative distribution of *AM* (according to data from [22, 23]) are shown in Table 1.

It can be concluded from Fig 1 and Table 1 that throughout life there is a complete cessation of hematopoiesis in some individual sites and gradual conversion of *AM* into *IM* in most of

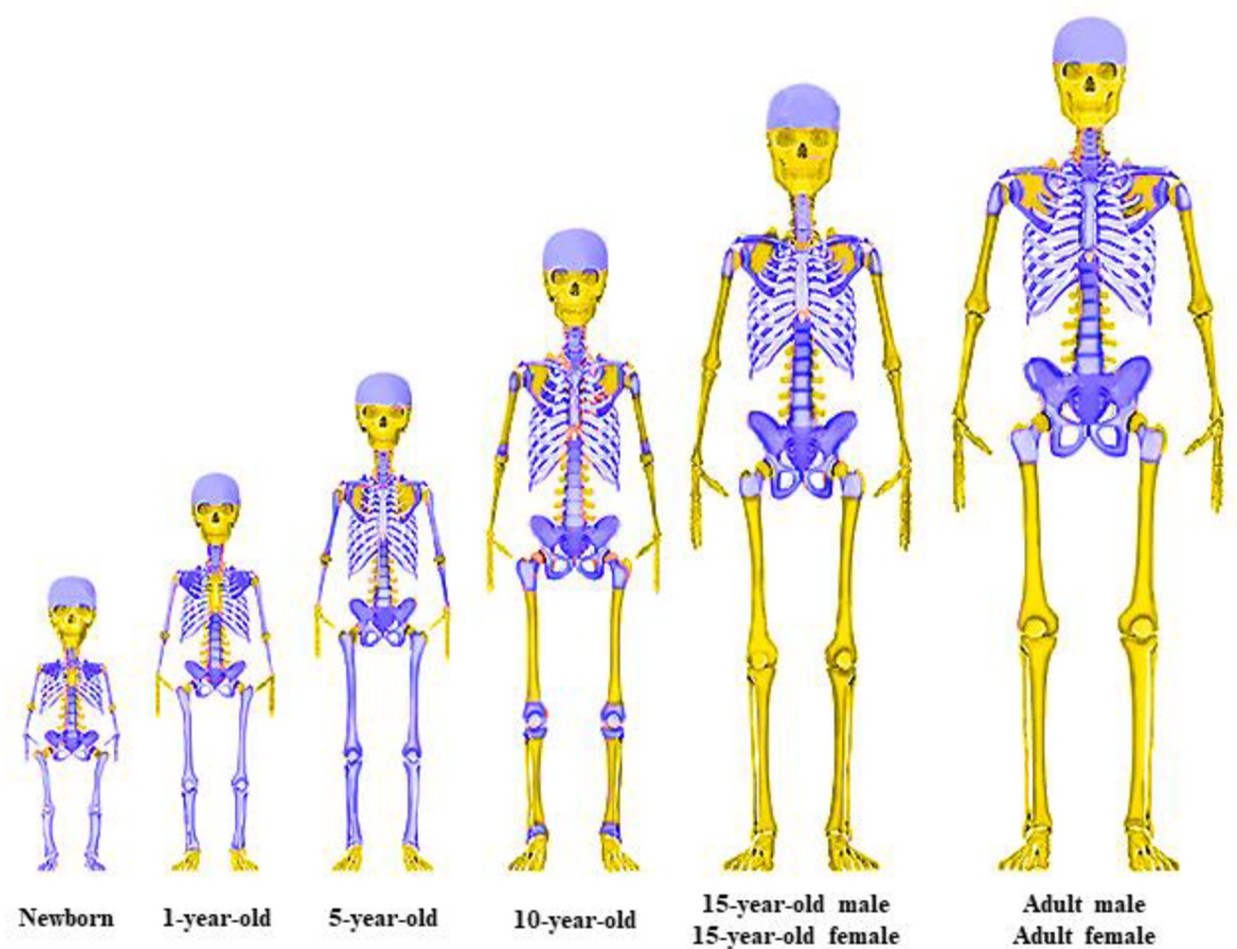

**Fig 1. Location of hematopoietic bone sites (purple color) in humans of different ages.** Other bones (yellow) do not contain active marrow and are not considered for modeling.

the remaining sites. For example, rapid cessation occurs during the first years of life of hematopoiesis in the fingers and toes, as well as in the middle of the diaphysis of long tubular bones. It should be noted that the timing of the replacement of hematopoietic *AM* with *IM* in various skeleton bones can vary considerably.

To analyze the data on quantitative distribution of *AM* (Table 1), it is important to note the methods used to obtain this data. Estimates from [22] were based on a collection of histologic data on age-specific marrow cellularity for specific bones, together with information on relative volumes of body regions (e.g., head, trunk, upper and lower legs) at the indicated ages. The contemporary method of positron-emission tomography with fluorothymidine as a label (FLT-PET) allows for *in vivo* visualization of the location and quantification of proliferating tissues, including *AM* [23]. The FLT-PET method made it possible to identify *in vivo* areas with the most pronounced proliferative activity of the marrow and assess the variability of proliferative activity in individual hematopoietic sites in a large group of adults [23]. Unfortunately, we did not find FLT-PET results for children (presumably, such data may appear in the coming years).

It can be seen from Table 1 that in newborns the largest fraction of *AM* is in the skull (28%), followed by the spine (14%), pelvis (11%), and wrists+hands+ankles+feet (11%). In

**Table 1. Main hematopoietic sites of human skeleton in different ages and distribution of *AM* for newborns and pre-adults [22] and for adults [23].**

| Bone site | Bone parts | *AM* (% from whole-skeleton *AM* mass) | | | | | |
|---|---|---|---|---|---|---|---|
| | | Newborn | 1 y | 5 y | 10 y | 15 y | Adult (mean±SD) |
| **Skull** | Plate bones, clivus | 28.2 | 28.7 | 18.1 | 12.8 | 10.2 | 6.2 ± 2.3 |
| **Ribs + Scapulae + Clavicles** | Central part of scapula | 10.1 | 11.8 | | | | |
| | Other parts | | | 12.8 | 14.8 | 18.0 | 15.3 ± 2.6 |
| **Vertebrae** | Cervical | 1.7 | 2.1 | 2.3 | 2.7 | 3.3 | 3.5 ± 1.0 |
| | Thoracic | 7.2 | 8.3 | 9.2 | 11.0 | 13.8 | 17.5 ± 2.4 |
| | Lumbar | 5.5 | 6.4 | 7.0 | 8.5 | 10.6 | 15.5 ± 2.5 |
| **Sacrum** | All parts | 4.4 | 5.1 | 5.7 | 6.8 | 8.5 | 7.4 ± 1.8 |
| **Pelvis** | Ilium, pubis, ischium | 11.4 | 13.1 | 13.5 | 15.9 | 18.6 | 23.2 ± 3.0 |
| **Femora** | Diaphysis | 6.7 | 8.1 | 13.5 | | | |
| | Distal parts | | | | 15.7 | | |
| | Proximal parts | | | | | 11.3 | 5.9 ± 2.5 |
| **Humeri** | Diaphysis | 4.5 | 5.2 | 4.8 | | | |
| | Distal part | | | | 4.1 | | |
| | Proximal parts | | | | | 3.8 | 3.6 ± 1.9 |
| **Tibiae and fibulae** | Diaphysis | 7.1 | 8.7 | 9.3 | | | |
| | Proximal and distal parts | | | | 5.6 | | |
| **Radii and ulnae** | All parts | 2.4 | 2.6 | 2.1 | | | |
| **Wrists and hands, Ankles and feet** | All parts | 10.8 | | | | | |

Note: shaded cells indicate active hematopoiesis (presence of *AM*), white cells indicate the absence or very low hematopoiesis (not considered for modeling).

adults, the greatest amount of *AM* is concentrated in the spine (36%), followed by pelvis (23%) and ribs+scapulae+clavicles (15%). The different sites have trabecular thicknesses and sizes of marrow cavities that are both site-specific and age-dependent.

## Bony structure in hematopoietical sites

Trabecular (cancellous or spongy) bone is a continuous network consisting of curved plates and rod-like elements intersecting with one another (Fig 2A). Basic parameters for description of bony structure are the micro-parameters trabecular thickness (*Tb.Th)*, trabecular separation or distance between trabeculae (*Tb.Sp*), and bone volume fraction of spongiosa (*BV/TV*); and the macro-parameters cortical thickness (*Ct.Th)* and overall dimensions. The abbreviations of the parameters are given in accordance with standardized nomenclature for bone histomorphometry [24, 25]. Different methods were used to evaluate the parameters, each of which has its pros and cons [26, 27]. A brief evaluation of the methods is given below.

Fig 2 illustrates an example of classical histomorphometric data obtained on a collection of histological slides prepared from iliac crest specimens of 67 adult humans of both genders [28]. In this late 1960s study, the parameters of more than 15 thousand trabeculae were measured under a microscope, and statistical distributions of trabecular thickness and trabecular separation were obtained. As can be seen from Fig 2C, both distributions were similar to log-normal, but mean trabecular thickness was significantly less than trabecular separation.

In the early 1980s, a semi-automated system for assessing bone parameters was proposed in [29]. The stereological basis of bone histomorphometry with the use of a theory of quantitative microscopy and reconstruction of the 3rd dimension was developed. The techniques were based on assumed parallel plate or rod trabecular models and isotropy of bone structures. The

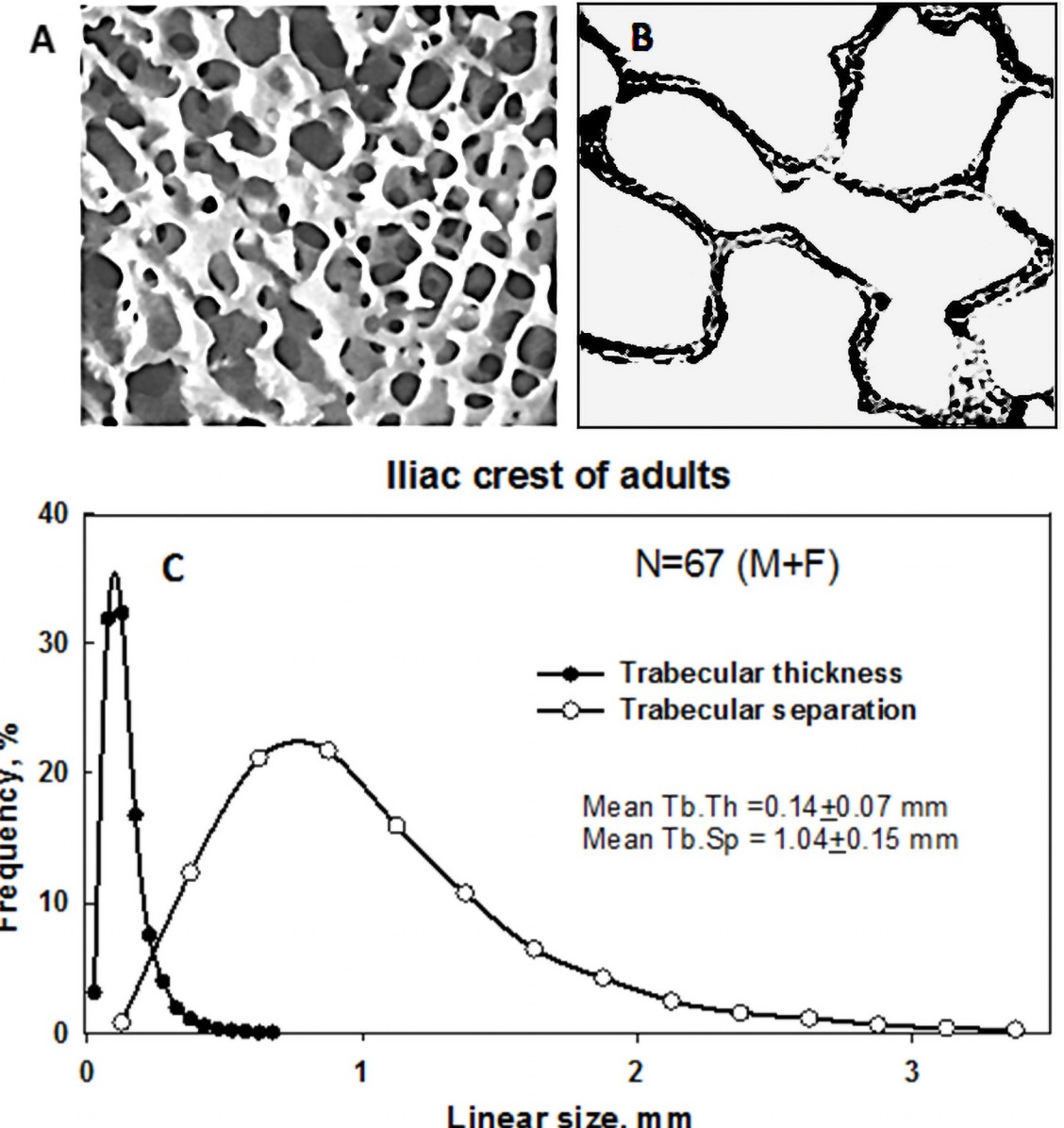

**Fig 2.** Spongiosa microstructure: (A) General view of spongy bone; (B) View of histological cross-section of spongy bone under a microscope: trabeculae are dark-colored and intertrabecular cavities are light-colored; (C) Statistical distributions of trabecular thickness and trabecular separation measured on a collection of histological slides prepared from iliac crest specimens based on data published in [28].

author noted that although his method is oversimplified, it is easy to use and can provide quick standardized results in different laboratories [30].

Since the 1990s, microcomputed tomography (microCT) has been used along with conventional histomorphometry to assess morphometric parameters of 3D trabecular structures [31–34]. The method allows for the study of a larger volume of spongiosa samples, and a quick evaluation of parameters (regardless of the rod or plate trabecular model); however, the measurement results significantly depend on microCT resolution [35, 36]. It should be noted that

some comparative studies of the histomorphometry and micro-CT measurements have shown high agreement [33, 34, 37], whereas others have reported only moderate agreement [31, 38, 39]. The difference may be attributable to inadequate resolution of microCT images relative to real trabecular size, the use of a plate trabecular model in 2D histomorphometry versus direct 3D methods in microCT, or other reasons. Chappard et al. [26] note that the estimates of *Tb. Th* depend on the three-dimensional reconstruction method used in histomorphometry. The highest agreement is observed for *BV/TV* estimates (the bias between the methods does not exceed 3%).

In the current study both microCT imaging with adequate resolution ($\leq 40$ μm) as well as histomorphometry data obtained using the rod trabecular model are considered for the assessment of bone parameters. Table 2 shows examples of morphometric parameters estimated by different authors for the vertebrae of adults, children, and newborns.

As shown in Table 2, there are numerous studies of adults, with sample means having a significant difference due to individual variability. Much less data has been published for newborns and young children, and the samples can be as small as one cadaver. In all available child studies, *BV/TV* was measured, but only two studies measured *Tb.Th*. Measured data on vertebra-cortical thickness for children were not found. However, bone images available in [49] show that cortical thickness of newborn- and infant-vertebrae is very thin and comparable to *Tb.Th*. Based on these images it was considered that the entire vertebral body in the early ages consists of trabecular bone only. Also, it should be noted that in the case of data paucity, relative standard uncertainties for the parameter values may be extrapolated from the corresponding data for other ages.

A distinction should be made between individual variability (which characterizes the differences between individuals) and intra-specimen variability (which characterizes the variation in the parameters for a particular person). Data on intra-specimen variability for bone sites are available in [50–52].

## Density and elemental composition of bone and marrow

The mineralized bone matrix has two major components: an organic matrix (mainly collagen) and bone minerals (hydroxyapatite, amorphous calcium phosphate and some other salts). Notably, the mineralized bone in living organisms contains the bone matrix with blood vessels, blood, nerves, fluid in lacunae and canals, as well as bound water on the surfaces of the bone matrix. All these components constitute hydrated (wet) mineralized bone. The density of such hydrated mineralized bone (i.e., tissue bone density) is considered here.

Estimates of tissue bone density for different ages are shown in Table 3. There are few publications on direct measurements of tissue bone density in children [53]. The data on indirect factors influencing the bone density (e.g., bone porosity) indicate significant differences for children of the first year of life vs other age-groups. The differences between children over 5 years old and adults usually do not exceed 5% (Table 3). The density of *AM* according to Tissue Properties Database [54] is equal to $1.029 \pm 0.002$ g cm$^{-3}$, which is very close to the density of *IM* of 0.98 g cm$^{-3}$ (SD not available). Assuming marrow density does not vary much by age, the marrow density is about half that of bone density for adults, and about 60% for newborns.

Elemental composition of mineralized bone of newborn and adult, and also composition of active and inactive marrow (*AM* and *IM*) of humans of all ages, is given in Table 4 [60].

As can be seen, the elemental composition of the mineralized bone for newborns is slightly different than for adults. Nevertheless, for both age groups the greatest contribution is made by oxygen, calcium and carbon. As for marrow, the largest contributors are carbon, oxygen and hydrogen both for *AM* and *IM*.

**Table 2. Examples of morphometric parameters estimated in different studies for the vertebrae of adults, children, and newborns (means and standard deviations).**

| Specimen | Number of subjects | *Tb.Th*, mm | *Tb.Sp*, mm | *Ct.Th*, mm | BV/TV, % | Source of data |
|---|---|---|---|---|---|---|
| **Lumbar vertebrae of adult** | 14 | 0.14±0.01 | 0.57±0.12 | - | 21±5 | [40] |
| | 9 | 0.12±0.01 | 0.72±0.08 | - | 16±2 | [41] |
| | 30 | 0.23±0.02 | 0.54±0.13 | - | 29±5 | [42] |
| | 7 | 0.13±0.03 | - | - | 15±4 | [43] |
| | 27 | 0.09±0.03 | - | 0.52±0.03 | - | [44] |
| | 4 | - | - | 0.49±0.05 | - | [45] |
| | 26 | - | - | 0.29±0.02 | - | [46] |
| **Vertebrae of newborn** | 1 | 0.25±0.01 | - | - | 64±3 | [47] |
| | 1 | 0.18±0.04 | - | - | 30±6 | [47] |
| | 1 | - | - | - | 60±9 | [48] |
| | 1 | - | - | - | 46±5 | [48] |
| **Vertebrae of children 0.5–9 years** | 1 | 0.17±0.02 | - | - | 20±4 | [47] |
| | 1 | 0.19±0.03 | - | - | 20±5 | [47] |
| | 15 | 0.09±0.01 | 0.60±0.12 | - | 13±3 | [49] |

## Specification of the dosimetry problem for beta-emitters

Strontium isotopes uniformly distributed in trabecular and cortical bone volumes (*TBV* and *CBV*) are considered to be source regions of ionizing radiation. The marrow-filled cavities surrounding the trabecular bone are considered to be the target region. It is assumed that *AM* cells are randomly distributed within the cavities, and absorbed doses are equal for *AM* and *IM*. The cortical bone volume (*CBV*) is considered as an additional exposure source for *AM*. Thus, the dosimetry problem is to evaluate energy deposition in target regions for particles emitted in *TBV* and *CBV* sources. This problem can be solved by stochastic modeling of radiation transport in a porous structure simulating the spatial distribution of bone and marrow within spongiosa. For practical application, skeletal-averaged dose factors *DF(AM←TBV)* and *DF(AM←CBV)* should be calculated, which represent dose rate in the marrow per unit concentration of $^{89,90}$Sr in *TBV* and *CBV* taking into account the *AM* fractions within the hematopoietic sites.

Energies of electrons from $^{89}$Sr spectrum are in the range of 0–1.5 MeV and the energy range for combined spectra of $^{90}$Sr/$^{90}$Y is 0–2.4 MeV. Average energy values in the two spectra are close to 0.6 MeV. Assuming the continuous-slowing-down approximation, the mean free path of electrons in spongiosa is about 2 mm, and greater than 99% of the energy would be absorbed within 10 mm. The thickness of trabeculae varies within the range of 0.04–0.4 mm, and for most emitted electrons such thin structures are penetrable. Thus, the geometric shape of the trabeculae does not play a significant role in the process of multiple scattering (on bone media) with respect to angular deflections and energy loss straggling. On the other hand, the

**Table 3. Estimates of tissue bone density in different ages.**

| Age range, years | Number of subjects | Tissue bone density, M ± SD or (range), g cm$^{-3}$ | Source of data |
|---|---|---|---|
| Newborns | N/A | 1.65 (1.5–1.8) | [55] |
| 2–19 | 14 | 1.84±0.06 | [56] |
| 20–75 | 25 | 1.91±0.05 | [57] |
| 29–73 | 7 | 1.86±0.06 | [58] |
| 38–75 | 10 | 1.87±0.03 | [59] |

**Table 4. Elemental composition of mineralized bone of newborns and adults and also active and inactive marrow of humans in all ages (% by mass) according to [60].**

| Element | Mineralized bone | | Marrow | |
|---|---|---|---|---|
| | Newborn | Adult | Active | Inactive |
| H | 4.2 | 3.5 | 10.5 | 11.5 |
| C | 16 | 16 | 41.4 | 64.4 |
| N | 4.5 | 4.2 | 3.4 | 0.7 |
| O | 50.2 | 44.5 | 43.9 | 23.1 |
| Na | 0 | 0.3 | 0.1 | 0.1 |
| Mg | 0.3 | 0.2 | 0.2 | 0 |
| P | 8.0 | 9.5 | 0.2 | 0.1 |
| S | 0.3 | 0.3 | 0.2 | 0.1 |
| Ca | 16.5 | 21.5 | 0 | 0 |
| Fe | 0 | 0 | 0.1 | 0 |

maximum path length in spongiosa is comparable to the linear dimensions of small bones and some electrons escape to cortical bone. To account for energy losses, the linear dimensions of bones must be considered in the dosimetry model.

## General approach and modeling tasks to be solved

A general approach to bone dosimetric modeling for $^{89}$Sr and $^{90}$Sr+$^{90}$Y was described in [61]. The hematopoietic sites are modeled by dividing them into small segments described by simple geometric shapes [20, 21]. The cortical layer with a thickness of *Ct.Th* is located on the outer (not adjacent to each other) walls of the segments. Spongiosa, which fills the segments, is modeled as an isotropic three-dimensional grid (framework) of rod-like trabeculae that "run through" the marrow. In the skeletal model, multiple framework deformations are simulated by changing the positions of the grid nodes and the thickness of the rods [21]. Model grid parameters are selected in accordance with the parameters of spongiosa microstructures taken from the published literature (*Tb.Th; Tb.Sp; BV/TV)*. Stochastic modeling of radiation transport in a heterogeneous medium simulating the distribution of bone tissue and marrow in each of the segments is performed by Monte Carlo methods [20].

The suggested approach to model development is based on completing several important tasks. A brief description of the essence of the proposed solutions are given below.

## Segmentation of hematopoietic sites

The first task includes collection and analysis of published data on macro-and micro-structure of the bone sites; segmentation of each bone site and segment description by simple geometric figures (e.g., boxes, regular and deformed cylinders, ellipsoids, etc.). An approach to segmentation of hematopoietic sites for calculations of dose to *AM* exposed to bone-seeking radionuclides was described in [62]. Stylized models of bone segments were created and their parameter values evaluated for fifteen hematopoietic sites (femur, humerus, fibula and tibia, radius and ulna, pelvis, ribs, vertebrae [cervical, lumbar, thoracic], sacrum, skull, clavicle, scapula, sternum, and hand and foot bones). The number of hematopoietic sites in an adult reduces to twelve with the removal of fibula and tibia, radius and ulna, and hand and foot bones. Sex-differences in macro-parameters of the sites were considered for people starting in adolescence. Some segmentation information is readily available in our previous publications. For example, segmentation of the vertebra and ribs of adult males were described in [20, 62].

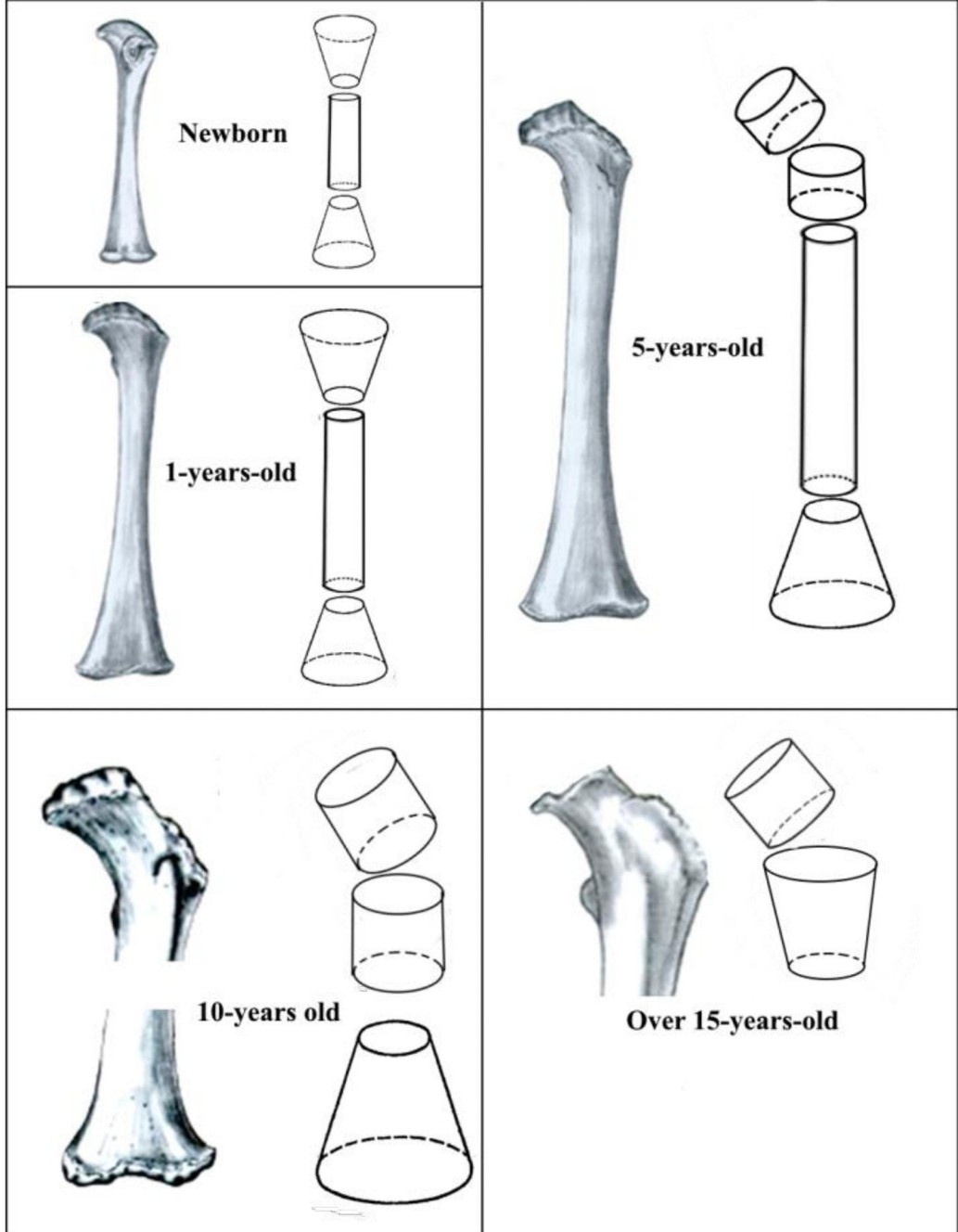

**Fig 3. Example of segmentation: Human femur as hematopoietic site at different ages and stylized segments assumed for its description.**

Femur modeling details are provided here to exemplify the process and details considered. Fig 3 shows the real forms of the hematopoietic human femur at different ages and their stylized models. As can be seen, from birth to 5 years old, a whole femur was modeled using three stylized segments. For a ten-year-old child, only the proximal and distal parts were modeled, because the entire marrow is inactive in the middle part of the bone. For ages over 15 years,

hematopoiesis occurs only in the proximal femur, the shape of which is modeled by two stylized segments (Fig 3).

It is important to note that epiphysis (secondary centers of ossification, i.e. femoral head and trochanteric epiphysis) were not included in the dosimetric model because in newborns they are mainly expressed as cartilage. When ossification begins in these areas, *AM* is rapidly replaced by *IM* [63–65].

Mean values of macro-parameters of the femur segments evaluated based upon published data [66–70] are shown in Table 5. Shafts and proximal ends of the femur were described by round cylinders (RC) of height *h* and diameter $d_1$; distal ends and trochanter area were described by deformed cylinders (DC) of height *h*, a round base of diameter $d_1$ and an elliptical base of axes $d_2$ and $d_3$; the neck was described by an elliptical cylinder (EC) of height *h* and of axes $d_1$ and $d_2$. The cortical layer thickness (*Ct.Th*) located on the outside walls of the segments was estimated based upon the data taken from [71, 72].

The changes in the linear dimensions of the segments reflect the processes of growth and development of the femur. As follows from Table 5 and Fig 3, there is an increase in the length of the femoral shaft (considered in the model for ages 0–5 years), as well as an increase in the transverse dimensions, that is, the diameters of various parts of the femur. The *Ct.Th* values generally increase with age. It is assumed that the growth of the femur is completed by the age of 15 years, and then the macro-parameters remain unchanged. Estimates of individual variability (in terms of CV) are in the ranges from 4% to 23% for the bone segment sizes and from 7% to 25% for the *Ct.Th* (Table 5).

The mean values of micro-parameters of the femur segments evaluated based upon published data [48, 69, 73–75] are shown in Table 6, including bone volume fraction of spongiosa (*BV/TV*); trabecula thickness (*Tb.Th*); and separation distance between trabeculae (*Tb.Sp*). Analysis of the published data has shown that the microstructure parameters do not depend on sex; therefore, data for men and women were combined for modeling.

As follows from Table 6, in childhood, an increase in both *Tb.Th* and *Tb.Sp* is observed. However in adults, the average thickness of trabeculae decreases and *Tb.Sp* keeps increasing. This results in a decrease of the proportion of bone in the total volume of spongiosa (*BV/TV*).

Estimates of individual variability (in terms of CV) can change from 9% to 22% for *Tb.Th* and from 13% to 27% for *Tb.Sp* depending on the data available for the specific bone segment (Table 6). Ranges of *BV/TV* tend to decrease with age. In general, we can say that the variability of adults' microstructure parameters with the fully formed skeleton is slightly lower compared with children whose skeleton continues to form.

## Stochastic generation of bone segment phantoms

The next task is the elaboration of a generator of 3D microstructures imitating the geometry and microstructure of bone segments in hematopoietic sites to produce multiple models of spongiosa. The stages of model generation were described in [20, 76]. In the first stage, the mathematical model of infinite spongiosa is generated. The mathematical basis for this was described in [21]. The spongiosa microarchitecture is simulated with interconnected rod-like structures representing trabeculae. Fig 4A illustrates a model example of four interconnected trabeculae. The spongiosa model results from deformation of a three-dimensional grid by stochastic node perturbation and random variation of the node thickness. The rod-like trabeculae, now simulated with cone segments resulting from varying end-node thicknesses, are either truncated along the edges of the deformed grid or connected with other trabeculae within the deformed grid using spheres [20, 21].

**Table 5. Femur segment macro-parameters and their individual variability estimated for different ages.**

| Age | Segment | Shape | Macro-parameters: mean, cm (CV, %) | | | | |
|---|---|---|---|---|---|---|---|
| | | | $h$ | $d_1$ | $d_2$ | $d_3$ | $Ct.Th$ |
| **Newborn** | Diaphysis | RC [(1)] | 3.78 (5) | 0.72 (11) | - | - | 0.17 (24) |
| | Proximal end | DC [(2)] | 1.89 (5) | 0.72 (11) | 2.64 (9) | 1.16 (12) | 0.05 (24) |
| | Distal end | DC | 1.89 (5) | 0.72 (11) | 2.64 (9) | 1.16 (12) | 0.04 (25) |
| **1 year** | Diaphysis | RC | 7.15 (4) | 1.12 (7) | - | - | 0.23 (17) |
| | Proximal end | DC | 3.58 (4) | 1.12 (7) | 3.4 (12) | 1.8 (8) | 0.07 (17) |
| | Distal end | DC | 3.58 (4) | 1.12 (7) | 3.4 (12) | 1.8 (8) | 0.06 (17) |
| **5 years** | Diaphysis | RC | 14.8 (4) | 1.66 (6) | - | - | 0.37 (8) |
| | Upper proximal end | RC | 2.47 (4) | 2.3 (23) | - | - | 0.13 (14) |
| | Lower proximal end | RC | 2.47 (4) | 2.3 (23) | - | - | 0.13 (14) |
| | Distal end | DC | 4.94 (4) | 1.66 (6) | 6.82 (6) | 2.5 (7) | 0.11 (7) |
| **10 years** | Upper proximal end | RC | 3.46 (5) | 2.49 (7) | - | - | 0.18 (17) |
| | Lower proximal end | RC | 3.46 (5) | 2.49 (7) | - | - | 0.22 (12) |
| | Distal end | DC | 6.92 (5) | 7.84 (7) | 3.32 (10) | 2.1 (9) | 0.11 (14) |
| **15 years Male** | Neck | EC [(3)] | 2.96 (5) | 3.6 (14) | 3.2 (13) | - | 0.20 (19) |
| **Adult Male** | Trochanter area | DC | 4.1 (4) | 6.6 (6) | 4.4 (6) | 3.0 (7) | 0.23 (15) |
| **15 years Female Adult Female** | Neck | EC | 3.05 (5) | 2.94 (10) | 2.39 (9) | - | 0.20 (19) |
| | Trochanter area | DC | 3.4 (5) | 5.8 (7) | 3.9 (7) | 2.7 (6) | 0.23 (15) |

[1.] RC–round cylinder.

[2.] DC–deformed cylinder.

[3.] EC–elliptical cylinder.

The mean values of *Tb.Th* and *Tb.Sp* and their corresponding standard deviations evaluated based upon published data are used to simulate intra-specimen variability of spongiosa microstructure (Fig 4A). Generated *Tb.Sp$^g$* is calibrated to fit the model to literature-derived *BV/TV* [76].

Then, a certain part of the modeled spongiosa is cut out in such a way as to fill the volume of the stylized bone segment and is voxelized. At this stage some rods can become "disconnected". At the last stage, a cortical layer of a given thickness is formed by replacing the outer layers of the spongiosa voxels with the voxels of the cortical bone. As a result, some trabecular rods contact with the cortex and other do not in a region intersecting with the cortical shell. Fig 4B presents the cross-sections of generated and voxelized models of three segments from vertebra, femur and iliac crest (on the same scale).

**Table 6. Femur trabecular micro-structure parameters and their individual variability estimated for different ages.**

| Age | Segment | BV/TV (min-max), % | Tb.Th, mean, mm (CV, %) | Tb.Sp, mean, mm (CV, %) |
|---|---|---|---|---|
| **Newborn** | All | 37 (9.6–53) | 0.11 (15) | 0.39 (27) |
| **1 year** | All | 22 (15–32) | 0.16 (22) | 0.54 (20) |
| **5–10 years** | Diaphysis and distal end | 26 (17–32) | 0.24 (22) | 0.54 (14) |
| | Proximal ends | 35 (23–43) | | |
| **15 years (M+F)** | Neck | 35 (23–43) | 0.24 (22) | 0.54 (14) |
| | Trochanter area | 26 (17–32) | | |
| **Adult** | Neck | 17 (14–22) | 0.19 (11) | 0.78 (13) |
| **(M+F)** | Trochanter area | 11 (8–13) | 0.14 (9) | 1.0 (13) |

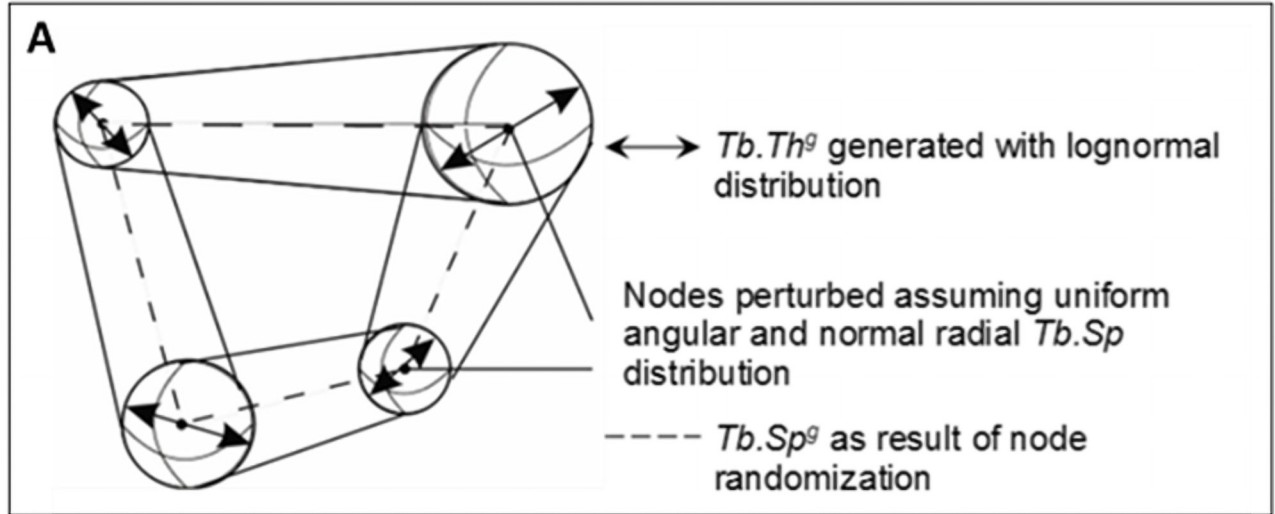

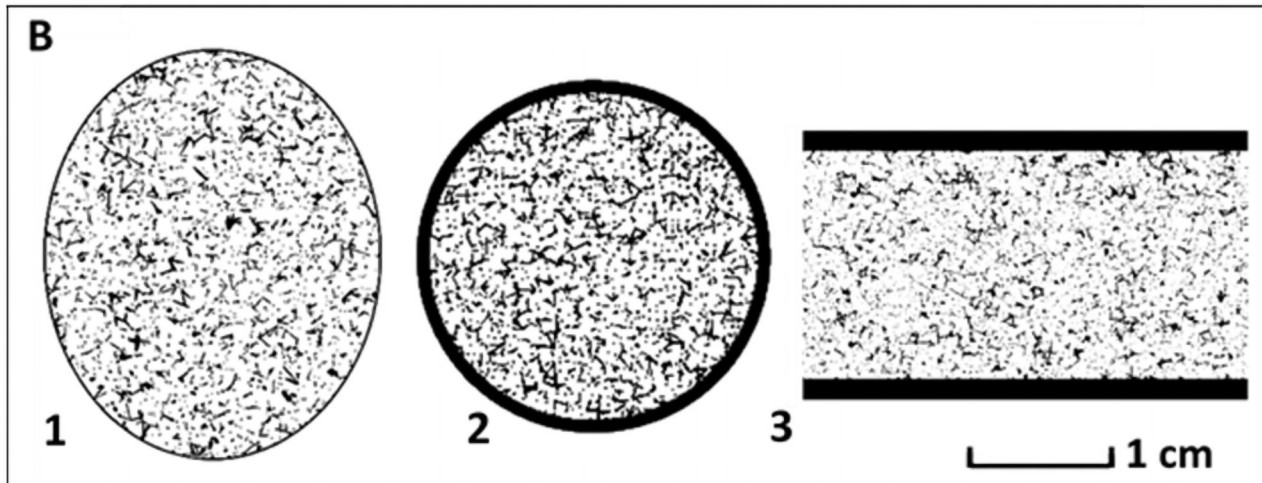

**Fig 4. Model examples of the generated trabecular structures.** (A) Scheme of four interconnected trabeculae and randomization of the grid and simulation of $Tb.Th^g$ and $Tb.Sp^g$ ($g$ indicates the generated output against the original input data). (B) Cross-sections of the voxelized models for three bone segments with different micro-parameters and cortical thickness (bone is black-colored and marrow is white-colored). (1) Thoracic vertebrae body in age = 10 y ($Ct.Th$ = 0.2 mm; $Tb.Th$ = 0.12 mm; $BV/TV$ = 0.13); (2) Femur lower proximal end in age = 5 y ($Ct.Th$ = 1.3 mm; $Tb.Th$ = 0.24 mm; $BV/TV$ = 0.35); (3) Iliac dorsal segment equal for adult male and female ($Ct.Th$ = 1.5 mm; $Tb.Th$ = 0.1 mm; $BV/TV$ = 0.2).

The original software "Trabecula" has been developed to simulate 3D bone segments in voxel presentation [76]. The software can perform random generation of trabecular bone structures inscribed into simplified bone shapes (e.g., boxes, regular and deformed cylinders, ellipsoids) that are covered by a cortical bone layer. The homogeneous cortical layer is generated in the process of voxelization (in particular, voxel resolution of femoral segment phantoms varies from 70 μm up to 160 μm, see S1 File). The voxel belonging to a particular structure (cortical bone, trabecular bone, marrow, or void) was defined according to the central point of the voxel [76]. The synthetic model of spongiosa was calibrated to fit the main input parameters after voxelization. Thus, the values of $Ct.Th$, $Tb.Th$ and $BV/TV$ of the voxelized models are equal to the input model parameters (see examples in S1 File). Volumes of the marrow target and the source-tissues ($CBV$ and $TBV$) are directly calculated by "Trabecula" for voxel phantoms.

The computer program supports dual functionality: generating a model with fixed parameters (phantom imitating the population average) and generating a set of phantoms imitating individual variability in the population. The random generation of spongiosa and model voxelization does not induce any significant changes (<1%) to the mean values of linear dimensions and tissue volumes of the segment-specific computational phantoms. By simulating the individual variability with 12 additional realizations of segment-specific phantoms, the variability of the parameters of the resulting model was estimated. Individual variability of the parameters of generated phantoms fully corresponds to the input parameters [76].

"Trabecula", the generator of bone segment geometry, is designed for construction of computational phantoms for subsequent dosimetric application, viz., for evaluation of energy deposition in marrow target per emitted particle. A beneficial feature of the "Trabecula" code is its ability to generate output files that are formatted as Monte Carlo N-Particle (MCNP) code [77] input as three lattices of voxels, namely, a whole computational phantom and two separate source tissues (cortical and trabecular bones).

## Stochastic modeling of radiation transport

The Monte Carlo calculations of the transport of electrons and secondary photons are done with the MCNP6 code [77], operated with continuous-energy nuclear and atomic data libraries such as the Evaluated Nuclear Data File (ENDF) system, the Evaluated Nuclear Data Library (ENDL), and the Activation Library (ACTL) compilations. The MCNP transport physics accurately account for the diffusion and slowing down of all radiations in the electron-photon cascade established in the media. The absorbed energy is scored throughout the marrow media by pulse-height tally providing analog Monte Carlo modeling. The output represents the energy deposition in the target region per emitted electron in the source region.

At first, the MCNP computations were performed for monoenergetic electrons in the range from 50 keV to 2 MeV, and the fractions of energy absorbed in the marrow from radiation sources uniformly distributed in *TBV* and *CBV* were calculated for three bone sites of an adult male (ribs, femur and vertebra). The energy dependencies were compared with published results [78] based upon direct data of micro-images of bone structures. The comparison showed a similarity in the energy dependencies which was a good Quality Assurance test for our model [61]. This confirmed that the generated phantoms are dosimetrically equivalent to real bone structures in the considered range of electron energies. Then the MCNP computations were performed for the beta spectra of $^{90}$Sr+$^{90}$Y and $^{89}$Sr.

## Preliminary modeling results: Femur as an example

Segment- and site-specific *DFs* calculated for human femur phantoms for the spectra of $^{90}$Sr +$^{90}$Y are presented in Table 7 (in units $10^{-14}$ Gy/s per Bq/kg). Estimates of individual variability (in terms of CV, %) are also presented in Table 7. A S1 File presents the data underlying the computational results described in Table 7 (input parameters and simulation results for average values and 12 random trials for each of the segments).

Site-average values were calculated from the segment-specific *DFs* weighted in accordance to fraction of *AM* that is contained in a particular segment relative to *AM* of the entire site. It should be noted that, for each age, the *AM* fraction in the total marrow within the hematopoietic site is considered the same for all segments. Use of this approach results in weighting factors that are proportional to segment-specific marrow volume. Segment-specific bone marrow volumes are calculated by the code "Trabecula".

The calculated marrow volumes are consistent with the knowledge that throughout human life a significant redistribution of *AM* occurs between different segments inside the femoral

**Table 7. Modeling results for human femur: *DF(AM←TBV)* and *DF(AM←CBV)* for $^{90}$Sr+$^{90}$Y spectrum and their individual variability in terms of CV.**

| Subject, age | Segment | Dose Factors, $10^{-14}$ Gy/s per Bq/kg (CV, %) | | | |
|---|---|---|---|---|---|
| | | *DF(AM←TBV)* | | *DF(AM←CBV)* | |
| | | Segment specific | Site-average | Segment specific | Site-average |
| **Newborn** | Diaphysis | 5.3 (19) | | 6.9 (17) | |
| | Proximal end | 6.9 (14) | 6.8 (15) | 1.2 (12) | 1.6 (15) |
| | Distal end | 7.0 (15) | | 1.0 (16) | |
| **1 year** | Diaphysis | 3.9 (29) | | 4.9 (16) | |
| | Proximal end | 4.6 (29) | 4.6 (27) | 1.2 (11) | 1.6 (13) |
| | Distal end | 4.8 (25) | | 1.1 (11) | |
| **5 years** | Diaphysis | 5.0 (21) | | 3.7 (12) | |
| | Upper proximal end | 7.1 (17) | | 1.3 (31) | |
| | Lower proximal end | 7.1 (18) | 5.5 (17) | 1.3 (21) | 1.5 (15) |
| | Distal end | 5.3 (18) | | 0.94 (12) | |
| **10 years** | Upper proximal end | 7.2 (21) | | 1.4 (13) | |
| | Lower proximal end | 7.2 (18) | 6.0 (16) | 1.5 (14) | 0.90 (12) |
| | Distal end | 5.6 (15) | | 0.73 (11) | |
| **15 years Male** | Neck | 7.3 (14) | 6.0 (11) | 0.94 (14) | 0.82 (10) |
| | Trochanter area | 5.5 (10) | | 0.78 (8) | |
| **15 years Female** | Neck | 7.3 (16) | 6.0 (17) | 1.3 (10) | 1.0 (8) |
| | Trochanter area | 5.5 (18) | | 0.90 (7) | |
| **Adult Male** | Neck | 4.1 (11) | 3.1 (12) | 1.1 (4) | 0.92 (12) |
| | Trochanter area | 2.7 (13) | | 0.87 (15) | |
| **Adult Female** | Neck | 4.1 (11) | 3.1 (13) | 1.4 (10) | 1.1 (11) |
| | Trochanter area | 2.7 (14) | | 0.99 (11) | |

site (see also Table 1). The *AM* fraction located in the diaphysis (shaft) of children aged ≤ 5 years does not exceed 10–18%. In the first year of life, the proximal and distal segments each contain about 45% of the available *AM*. Then the contribution of "distal *AM*" begins to increase and by the age of 10 it reaches 75%. After this age, hematopoiesis in the distal segment gradually decreases, and from the age of 15 years all *AM* is concentrated in the proximal segments, with 70–75% in the trochanter area. Thus, to analyze age-dependencies of site-average values of *DF* (Table 7), the parameters of the segments with the predominant location of the *AM* must be taken into account. Comparison of age dependencies of *DFs* from Table 7 with the parameter values given in Tables 5 and 6 shows features described in the following paragraphs.

*DF(AM←TBV)* values (Table 7) generally reflect age-related changes in *BV/TV* (Table 6). The maximum site-averaged value (6.9) was obtained for newborns, for which *BV/TV* was 37%. Intermediate values (4.6–6.0) were obtained for the age range of 1 to 15 years, in which BV/TV changes amounted to 22–35%. This is also superimposed on the fact that from 5 to 15 years, the thickness of the trabeculae has maximum values (*Tb.Th* = 0.24 mm). By adulthood, *DF(AM←TBV)* values decrease by half, due to the fact that the bone porosity increases significantly: *BV/TV* reduces to 11% and the distance between the trabeculae reaches the maximum value (*Tb.Sp* = 0.8–1.0 mm). Individual variability of site-averaged *DF(AM←TBV)* is in the range from 11% to 27% and has no pronounced age dependence (Table 7).

The maximum values of site-averaged *DF(AM←CBV)* range from 1.5–1.6 (Table 7) and are observed around the first years of life, when the size of the bone segments is small (Table 5). For this age, increased site-average values are observed due to *AM* cells located in the diaphysis (shaft), where the dose rate is 5–10 times higher than in the proximal and distal segments with

inverse relationship to *Ct.Th* (Table 7). From the age of 10 years, *DF(AM←CBV)* values stabilize, being in the range of 0.8–1.1 (Table 7). This indicates that further bone growth does not have a significant effect on exposure from cortical bone to spongiosa of the *AM* located in the proximal and distal segments of the femur. Individual variability of *DF(AM←CBV)* is in the range from 8% to 15% and has some age-dependence, being lower in the 10–15 years range (Table 7).

The overall uncertainty of *DFs* is comprised of the uncertainty associated with the biological variability of the object properties and the uncertainty associated with simplifications made for modeling purposes [79]. Table 7 demonstrates the examples of the segment-specific (and site-average) *DF* uncertainties due to individual variability of bone macro and microarchitecture in the femur.

Each hematopoietic site has its own specific age dependencies on the *DFs*. In general, the following parameters have a significant influence on cortical and trabecular *DFs*: (1) the size of the bone that determines the volume of spongiosa, (2) the characteristics of the spongiosa microstructure, mainly *BV/TV*, and (3) the thickness of the cortical layer.

Skeletal averaged *DF* values are calculated from the site-specific *DFs* weighted in accordance with the age-dependent fractions of *AM* shown in Table 1. At the same time, additional uncertainties associated with variability in *AM* distribution among the hematopoietic sites are introduced into the estimates.

## Uncertainty description

The uncertainty associated with the biological variability is determined by variabilities in macro and microarchitecture of bones, density and chemical composition of bone, as well as variability in *AM* distribution among the hematopoietic sites. The overall uncertainty includes a contribution associated with simplifications made for modeling purposes (stylization of bone geometry, voxel resolution and assumption on uniformity of cortical thickness).

Preliminary estimates of uncertainties [79] identified the main contributors to overall uncertainty of skeletal averaged *DF*, viz., the individual variability of bone macro and microarchitecture and individual variability of *AM* distribution among the bone sites. The benefit of the SPSD approach is its ability to simulate individual variability. Generation of the set of segment-specific phantoms with randomly varied dimensions (within the range of literature-derived uncertainties of the model parameters) allows for the creation of a set of supplementary phantoms in addition to the population-average (basic) one. The uncertainty due to individual variability of bone micro and macro dimensions can be estimated as relative root mean square deviation (RMSD) between the dose factors calculated for supplementary models and *DFs* calculated with the basic one. According to the preliminary estimates [79], the segment-specific uncertainties for all bone sites may vary from 5% to 60% (about 20% on average). Variability of site-specific *AM* fraction for adults (according to Table 1) is in the range of 15–65% (about 30% on average). The contributions of other sources of uncertainty are of less significance, not exceeding 15%. According to the preliminary estimates, the overall uncertainty of femoral site-average *DFs* presented in Table 7 varies from 13% to 32%. Thus, overall uncertainties of skeletal averaged *DFs* are preliminarily estimated to be 32–54% for *DF(AM←TBV)* and 38–62% for *DF(AM←CBV)*. Currently, a separate article is being prepared which will describe in detail the methods for assessing uncertainties and present the results of their assessment.

## Discussion and conclusions

To lead off the discussion, the reader is reminded of three important aspects of this study: (1) the motivation of the study was to improve estimates of radiation risk of leukemia in the Techa

River cohorts, where [90]Sr and [89]Sr were the major marrow dose contributors; (2) the proposed model was developed specifically for beta emitters with a relatively high spectrum-average electron energy of about 0.6 MeV; and (3) active marrow was considered to be the radiosensitive skeletal tissue (while bone endosteum was not considered here). Thus, it is not reasonable to compare the "specific" model suggested here with the "universal" image-based (voxel- and mesh-type) reference computational phantoms developed for ICRP [80, 81]. On the other hand, an advantage of the proposed approach is that it allows the use of large available datasets for the assessment of the parameters of bone structures accumulated in studies not related to radiation (e.g., histology, anthropology, treatment of osteoporosis, bone prosthetics). Thus, the suggested *parametric method* does not require data input from additional autopsy material. The method makes it possible to avoid the main limitation of the *image-based method*: the fact that its development and improvement directly links with the necessity of imaging additional cadavers.

The dose factors *DF(AM←TBV)* and *DF(AM←CBV)*, as illustrated with the femur information presented, provide direct conversion of [90]Sr and [89]Sr concentration in the source regions (*TBV* and *CBV)* to the dose rate in the target region (AM). At present, work is being completed on the analysis of age dependencies for a full set of site-specific *DFs* and the calculation of skeletal averaged *DFs* and their uncertainties. As illustrative examples of the application of the *DFs*, the estimates of the dose rate are presented for the femoral *AM* of Techa riverside residents who ingested a significant amount of [90]Sr from river water contaminated by uranium fission products in 1949–1956 [3]. The average skeletal estimates are expected to be similar to these specific values. Maximum lifetime intake of [90]Sr was estimated to be 2–3 MBq in the early 1950s [11]. Monitoring of [90]Sr-body burden in Urals residents has been ongoing for over 60 years [12]. The upper panel of Fig 5 demonstrates the results of long-term monitoring of average cohort [90]Sr-body burden in residents of the upper and middle Techa who were adults during the period of maximal [90]Sr intake [12].

Since 1951, [90]Sr-body burdens were estimated by post-mortem analysis of bone samples, and since 1974, *in vivo* measurements have been performed on a large scale using a specially designed whole-body counter [12, 13]. As can be seen from Fig 5, the [90]Sr-body burdens have decreased by an order of magnitude over 50 years. The corresponding [90]Sr concentrations in *TBV* and *CBV*, (middle panel of Fig 5) were estimated using the biokinetic model described in [15] based upon the intake scenario described in [11]. As can be seen from Fig 5, there was a significant redistribution of [90]Sr in the skeleton: if in the initial period [90]Sr was predominantly incorporated in *TBV*, then by the end of the observation period almost all radionuclide was concentrated in *CBV*. The lower panel of Fig 5 demonstrates dose rate absorbed in femoral *AM*, estimated using the dosimetric model based upon [90]Sr concentrations in *TBV* and *CBV*. As can be seen from Fig 5, the dose rate in *AM* has decreased by almost two orders of magnitude over 50 years of observation.

Whole-body counter measurements for different age groups of Techa residents were also published in [12, 14], which makes it possible to assess the age dependence of the dose rate in *AM* in the long-term period after the maximum intake (Fig 6). By the time that collection of whole-body measurements had been initiated, all people who received the largest intake had become adults and nearly all of the remaining [90]Sr was located in the *CBV*. As can be seen from Fig 6, the [90]Sr-body burdens and the dose rates are proportional and have a noticeable maximum for people who were in adolescence during the period of maximum intake (born in the mid-1930s). Maximum dose rates in this age group in 1980 were about 40 mGy/year then decreased to 5 mGy/year in 2010 (Fig 6).

It should be noted that the dose rates in the described examples are given only for ingested [90]Sr using femoral DFs. Reviewing Table 1 data, total femoral marrow dose is an age-

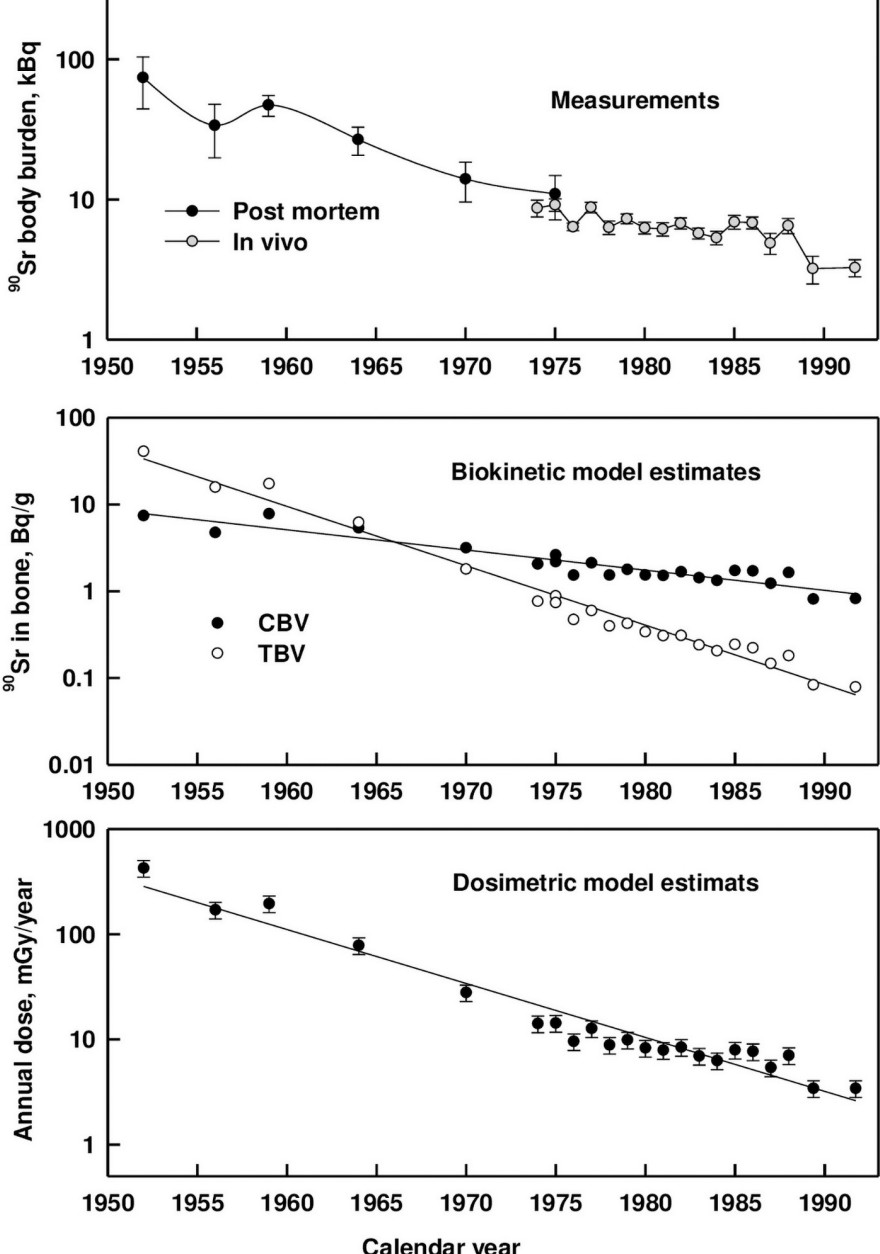

**Fig 5. Measured $^{90}$Sr body burden and modeled bone burden and dose for the adult residents of the Techa River.**
The upper panel shows average $^{90}$Sr measurements (vertical bars represent error of mean); the middle panel shows the corresponding $^{90}$Sr concentrations in *TBV* and *CBV*, estimated by the biokinetic model; the lower panel demonstrates dose rates absorbed in femoral *AM*, estimated by the dosimetric model.

dependent fraction of total skeletal marrow dose. In addition, the DF examples do not reflect the total dose rate in *AM*, which was also determined by the contributions from ingested $^{89}$Sr and $^{137}$Cs, as well as external exposure of the whole body [4]. Thus, for the future analysis of the excess relative risk of leukemia observed in the Techa River cohorts, together with the $^{90}$Sr doses shown here, the total doses from all exposure pathways will be estimated. Further study of these unique cohorts provides an opportunity to gain more in-depth knowledge about the

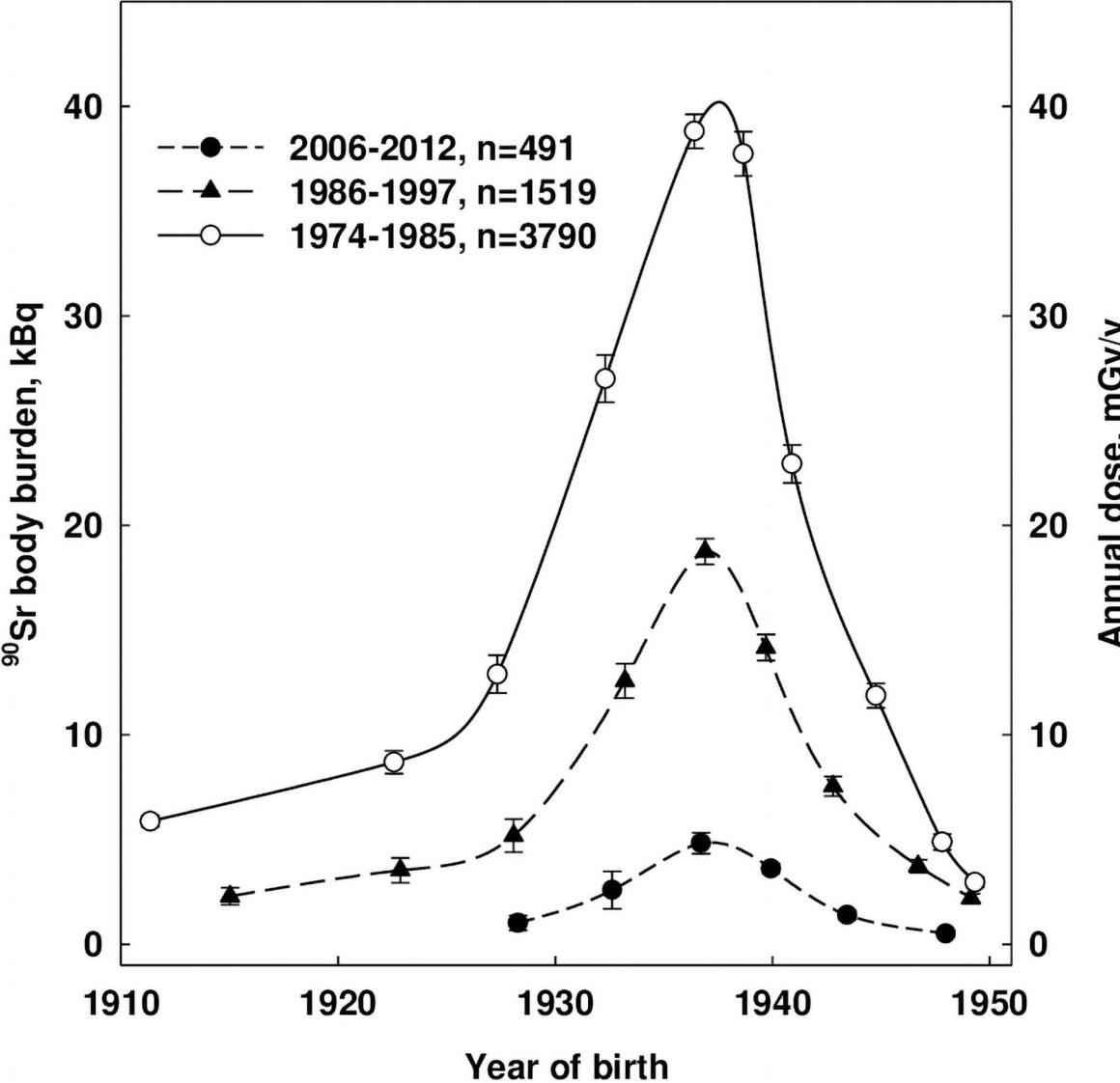

**Fig 6. Estimation of dose rate in femoral *AM* for different age cohorts of the Techa River residents with ⁹⁰Sr-body burden measured during 3 different time periods (vertical bars represent error of mean).**

effects of chronic radiation on the hematopoietic system. In addition, the proposed model can be used to assess the doses to *AM* under any other scenarios of ⁹⁰Sr and ⁸⁹Sr intake to humans.

Thus, a methodology of bone dosimetry modeling has been developed which includes parametric random generation of computational phantoms of bone segments for hematopoietic sites of human skeleton in voxel representation. The methodology for the calculation of doses absorbed in the marrow from beta-emitting bone-seeking radionuclides does not require additional studies of autopsy material. In this article, the formulations of the tasks and a brief description of the proposed approach are presented. Completion of these important tasks and the modeling results will be the subject of a separate article which is now under preparation. The modeling results will be incorporated into a special Monte Carlo dosimetry system that provides multiple realizations of individual dose estimates for each member of an epidemiological cohort [82]. Thus, confidence intervals in cancer risk estimates for the cohort can be

estimated by the radiation epidemiologists in accordance with the dosimetry uncertainties [83].

## Supporting information

**S1 File. The set of data supporting Table 7 describes the segment-specific computational phantoms of femur for humans of different age and gender (input and output parameters) as well as corresponding results of *DF(AM←TBV)* and *DF(AM←CBV)*.** The simulation results are given for average values and 12 random trials for each of the segments evaluated. The "terms and explanations" worksheet contains the explanations for the designations used in the tables with the modeling results.
(XLSX)

## Acknowledgments

The authors are sincerely grateful to Dr. Derek Jokisch (Francis Marion University) for helpful comments and discussions.

## Author Contributions

**Conceptualization:** Marina O. Degteva, Elena A. Shishkina, Vladimir I. Zalyapin.

**Formal analysis:** Marina O. Degteva, Evgenia I. Tolstykh, Elena A. Shishkina.

**Funding acquisition:** Bruce A. Napier.

**Investigation:** Pavel A. Sharagin, Alexandra Yu. Volchkova, Michael A. Smith.

**Methodology:** Evgenia I. Tolstykh.

**Project administration:** Bruce A. Napier.

**Software:** Pavel A. Sharagin.

**Supervision:** Bruce A. Napier.

**Writing – original draft:** Marina O. Degteva, Evgenia I. Tolstykh, Elena A. Shishkina, Bruce A. Napier.

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
