## [Decision Letter · Decision Letter 0]

15 Jul 2021

PONE-D-21-09605

Stochastic Parametric Skeletal Dosimetry model for humans: General description

PLOS ONE

Dear Dr. Napier,

Thank you for submitting your manuscript to PLOS ONE. After careful consideration, we feel that it has merit but does not fully meet PLOS ONE’s publication criteria as it currently stands. Therefore, we invite you to submit a revised version of the manuscript that addresses the points raised during the review process.

Both reviewers have identified a number of points that need to be clarified in order to improve the reproducibility of your study as reported. Please ensure you carefully respond to each of the items raised by the reviewers when preparing your revisions.

We look forward to receiving your revised manuscript.

Kind regards,

Jamie Males

Staff Editor

PLOS ONE

Journal Requirements:

Additional Editor Comments (if provided):

Reviewers' comments:

Reviewer's Responses to Questions

**Comments to the Author**

1. Is the manuscript technically sound, and do the data support the conclusions?

Reviewer #1: Yes

Reviewer #2: Yes

2. Has the statistical analysis been performed appropriately and rigorously? 

Reviewer #1: Yes

Reviewer #2: Yes

3. Have the authors made all data underlying the findings in their manuscript fully available?

Reviewer #1: No

Reviewer #2: Yes

4. Is the manuscript presented in an intelligible fashion and written in standard English?

Reviewer #1: Yes

Reviewer #2: Yes

5. Review Comments to the Author

Reviewer #1: The manuscript presents an approach for computing dose factors to active marrow from intakes of Sr-89 and Sr-90 in the human femur. The authors describe construction of a synthetic, digital femoral model which is then coupled to a Monte Carlo transport code to model electron energy deposition from strontium radionuclides in the bone volume. The paper is thorough and generally well described. I recommend revisions, several of which fall into the 'major' category. I opted for "minor revisions" overall. While I have a large number of comments and suggestions, I don't believe any necessitate repeating portions of the study.

Major comments

1. As described by the authors, the paper presents a dosimetry model and methodology applied to active marrow dose from Sr-89 and Sr-90 present in the human femur. The current title is, however, broader in scope. The authors should consider revising the title to specifically include mention of the femur and the two strontium radionuclides. They could also include mention of active marrow dose, since they acknowledge in the paper that the method does not provide dose estimates to the bone endosteum, an important component of "skeletal dosimetry" in radiation protection. It is certainly appropriate for the authors to write that the same methodology presented in the manuscript can be extended to other skeletal sites and other beta emitters, as they have done. But, the current manuscript is limited to a single skeletal site and two radionuclides.

2. Line 164: "In the current study both microCT imaging with adequate resolution..." For the sake of reproducibility, the authors should specify what specific resolution value was determined to be "adequate resolution". The authors could also consider providing a justification for the value they used in this study.

3. Lines 174-176: "Measured data on vertebra-cortical thickness for children were not found. However, bone images available in [49] show that cortical thickness of infant- and child- vertebra is very thin and comparable to Tb.Th." The authors should state how, or if, they applied this estimate of cortical thickness in the vertebra of children to the femur. I'm also curious how the lack of data for this important parameter folds in to the authors' uncertainty analysis.

4. The biokinetic model utilized (reference [15]) includes both bone volume and surface compartments. The energy deposition model used by the authors assumes uniform activity distribution throughout the bone volume. If significant Sr decays take place on the bone surface, the authors would have a problem. Presumably, the authors are okay with their approach because Sr removal rate from the bone surface compartments is fast and any resulting non-uniformity of activity throughout the bone volume is negligible. If true, the authors should consider describing this in the paper. It might also be helpful to describe the early times following intakes in children. In other words, how soon following an intake is the assumption of uniform Sr activity distribution in bone volume a valid one given the biokinetic model in [15]?

5. Lines 235-247 and lines 313-353: Are all the trabecular rods in the synthetic model connected? And, how do they interface with the cortical shell model? The latter could be particularly important for cortical bone volume sources. If available, an image of the synthetic model analogous to Figure 2a could be helpful to readers, particularly if the image could show a region intersecting with the cortical shell.

6. Did the authors attempt to measure the important skeletal parameters after voxelization of their synthetic model? In other words, were the skeletal size parameters (Tb.Th, Ct.Th, BV/TV, etc.) used to create the synthetic model retained after voxelization?

7. Line 367: "The energy dependencies were compared with published results [78] based upon direct data of micro-images of bone structures. The comparison showed a similarity in the energy dependencies which was a good Quality Assurance test for our model [61]. This confirmed that the generated phantoms are dosimetrically equivalent to real bone structures in the considered range of electron energies." It would be appropriate to provide a plot showing this comparison. The conclusion re: "dosimetrically equivalent" is difficult to make in a publication without the supporting evidence.

8. Lines 463-465: "...active marrow was considered to be the radiosensitive skeletal tissue (while bone endosteum was not considered here). Thus, it is not reasonable to compare the "specific" model suggested here with the "universal" image-based (voxel- and mesh-type) reference computational phantoms developed for ICRP [80,81]." The authors might consider expanding upon the endosteum comment. For example, why this target is not important in their application.

Comparisons between the authors work would not be made with ICPR mesh or voxel phantoms since they are not used to compute energy deposition in active marrow from bone volume sources. More appropriate would be comparisons to specific absorbed fractions for the adult published in ICRP Pub. 133 for bone volume sources irradiating active marrow. Or, a comparison with dose coefficients to the active marrow for Sr-89, 90 published in the ICRP OIR series. Given the authors concluding statements on lines 524-525 and 43-44, such comparisons seem desirable. The values above, of course, represent activity and doses in the whole skeleton, so could not be made with the femoral data presented in this work. But, the authors could compare to the femoral data in Hough et al (their ref [78]) which is the basis for the ICRP SAFs.

9. Lines 470-472, 529-530: Given the sparsity of data the authors found for skeletal size parameters in children, particularly young children, would the current study not also benefit from additional measurements in cadavers?

Minor comments

a. In the abstract (line 38), the authors write: "The biokinetic model results will be used to calculate individual doses to members of a cohort exposed to 89,90Sr from liquid radioactive waste..." The authors should consider rephrasing to limit this to femur doses or to place after a reference to 'further' or 'future study'. As currently written, a reader would reasonably expect to find skeletal-wide doses.

b. Introduction, third paragraph (lines 66-67): Please consider improving/rephrasing: "Strontium intakes are incorporated... and the Sr become a source of beta radiation for the marrow."

c. Introduction, third paragraph (line 70): "... so called "chord-based" methods..." While other papers have used the term 'chord', Darley pointed out the more appropriate term is 'path length' or 'path-length'. See Darley, Philip J. Health Physics. 90(2):176-177, February 2006 and the response to Darley by Shah, Amish P.; Rajon, Didier A.; Jokisch, Derek W.; Health Physics. 90(2):177-179, February 2006.

d. Introduction, third paragraph (line 71): Reference [18] does not describe radiation transport in a voxelized skeletal model. Reference [78] (Hough et al.) does and the authors might consider citing it, here. The authors should also consider citing voxel studies published by Kramer et al. and Gao et al.

ELECTRON ABSORBED FRACTIONS IN AN IMAGE-BASED MICROSCOPIC SKELETAL DOSIMETRY MODEL OF CHINESE ADULT MALE

Shenshen Gao, Li Ren, Rui Qiu, Zhen Wu, Chunyan Li, Junli Li

Radiation Protection Dosimetry, Volume 175, Issue 4, August 2017, Pages 450–459, https://doi.org/10.1093/rpd/ncw372

Skeletal dosimetry based on µCT images of trabecular bone: update and comparisons

R Kramer1, V F Cassola1, J W Vieira2,3, H J Khoury1, C A B de Oliveira Lira1 and K Robson Brown4

Published 7 June 2012 • © 2012 Institute of Physics and Engineering in Medicine

Physics in Medicine & Biology, Volume 57, Number 12

Citation R Kramer et al 2012 Phys. Med. Biol. 57 3995

e. Lines 83-84: "In the current study, the spongiosa contains both source areas (bone trabeculae) and target areas (active and inactive marrow)." I believe the authors intend for the source regions to be both cortical and trabecular bone. Similarly, inactive marrow is not the target of interest, though the authors later explain that they have approximated the dose to the entire marrow space (active and inactive) as being equivalent to the active marrow dose.

f. Line 130: "Different methods were used to evaluate the parameters, each of which has its pros and cons and is not completely accurate due to the complexity of the 3D bone structure." The authors should consider improving this sentence. It is not clear what basis was used to "evaluate the parameters" or what quantity is being referenced as "not completely accurate". The sentence is too generalized as currently written.

g. Line 161: "Chappard et al. note that ... depend on the three-dimensional reconstruction method used in histomorphometry (better agreement is observed for the rod trabecular models)." The authors may wish to revisit this sentence. I found the parenthetical confusing... better agreement to what? Perhaps the parenthetical is out of context and can be deleted? The authors may also wish to revisit the next sentence, ("At the same time...") as well.

h. Line 211: "The marrow-filled cavities... are considered to be the target regions." While the authors later describe that dose to the bone endosteum is not computed in this model, the authors should also consider pointing it out here, at first use.

i. Line 213: "...the density of AM and IM is almost the same..." The two densities are roughly 5% different. The authors should consider replacing 'almost the same' with something more specific about the difference, as they have been elsewhere in the paper when similar sized differences exist.

j. Line 247: The authors should consider replacing "dosimetric equivalence" with a different phrase or description which does not require quotation marks.

k. Line 261: "Sex-differences ... were identified for people over 15 years old." Does this mean sex differences only exist in the adult models? If so, the authors should consider rephrasing the sentence or adding that conclusion.

l. Line 320: The authors should consider improving the sentence which begins: "The rod-like trabeculae along the edges..."

m. Lines 385-388: The authors should consider rewriting these two sentences. Specifically, perhaps the order should be switched? e.g. The method is ... The code "Trabeculae" implements that method.

n. Line 389: "The calculated marrow volumes showed that..." The authors should consider rewriting this sentence to state that the model is consistent with AM redistribution throughout life. The model itself is not evidence of that redistribution.

o. Line 412: "...where the crossfire dose rate is..." The authors should consider defining what is meant by "crossfire dose rate".

p. Line 414: Add units to the dose factor values at the end of the sentence.

q. The authors should look out for multiple places in the text on and on figures where the dose to the marrow is specific to the femur only. Please consider adding the word "femur" in such locations so the reader does not get confused. The right axis on Figure 6 and the left axis on Figure 5-bottom seem particularly important places to add 'femur' or 'femoral'.

r. Line 506: Would "largest intake" be more appropriate than "maximum intake"?

s. Lines 520-522: Please consider rephrasing or removing the sentence beginning with "Thus, for the analysis of the excess..." This falls into the future work category, correct?

t. Line 214: "The cortical bone volume (CBV) is considered as a source of crossfire exposure of the marrow. Thus, the dosimetry problem is to evaluate energy deposition in target regions for particles emitted in TBV and CBV sources." Should the first sentence also include TBV? Should target "regions" be "region" since the authors consider one target (the marrow)?

Reviewer #2: PONE-D-21-9605

Stochastic Parametric Skeletal Dosimetry Model for Humans: General Description

Degteva et al.

General Comments

This submitted paper describes a new and innovative approach to skeletal dosimetry with specific application to bone-seeking beta-particle emitters. The application is general to all beta-emitter bone dosimetry assessments but the specific purpose for this model development is The approach – while initially simplistic in its description of bone morphometry – is upon further reflection quite elegant and appropriate to the application of active marrow dose assessment. The vast majority of skeletal dosimetry models over the decades have been based upon direct imaging of cadaveric skeletal samples – from 2D optical scanning of bone slice radiograph to 3D imaging by NMR microscopy and microCT scanning. The authors of this report have taken a different approach – to model both the skeletal macrostructure and microstructure via stylized computational approaches – simple geometric shapes. They argue – quite successfully – that the simplicity of the morphometric models does not detract from dose accuracy, but at the same time, these types of geometric models allow for a stochastic sampling of a number of dosimetrically-important features to include tissue macrostructural dimensions (such as the thickness of the bone site cortex of cortical bone), tissue mixtures (marrow cavities and bone trabeculae within spongiosa), elemental compositions, tissue compositions (such as the mixture of hematopoietically active versus inactive bone marrow), and tissue microstructural dimensions (such as the thicknesses of the bone trabeculae and marrow cavity spaces). With this ability to “change” the model, following detailed collection of literature values and their statistical uncertainties and ranges, a fully stochastic bone dosimetry model can be implemented. For a purely image-based modeling approach this would require a significant cost and time for multiple cadaveric tissue collections and imaging sessions. One limitation of the model is its application to the pediatric child due to a paucity of data on bone morphometry and tissue features – but this is clearly not the fault of the modelers, and they do the best they can with this limited data source.

The paper is very well-written, clearly organized, and highly referenced. I have no specific editorial changes to make.

I would like the authors however to comment on the following issues:

• The title of the article might include the phrase – “General descriptions and applications to the femur” as most of the dosimetry data presented here is limited to this one bone site.

• The authors should further justify why voxelized models are used in the final radiation transport simulations. Could not they present their model in a polygon mesh format with direct coupling to the transport code? This would fully eliminate any voxel-size effects in the dosimetric results.

• What geometric expressions are to be used for more complex bone sites such as the cranium, mandible, vertebrae, and pelvis?

• For high-energy beta emitters in bone, there will be bremsstrahlung x-ray production and cross-bone site irradiation. How will this be handled within the SPSD model?

• Can the authors make a brief comment on final dosimetry results feed to the radiation epidemiologists? What will they do with stochastically sampled active marrow bone doses? Still use only the central estimates of bone marrow dose or will the dosimetry uncertainties carry through to cancer risk uncertainties?

Specific Comments

None – very well written and informative paper. My congratulations to the authors.

6. PLOS authors have the option to publish the peer review history of their article (what does this mean?). If published, this will include your full peer review and any attached files.

Reviewer #1: **Yes: **Derek W. Jokisch

Reviewer #2: **Yes: **Wesley E. Bolch

---

## [Author Response · Author response to Decision Letter 0]

13 Aug 2021

Dear Editor,

Thank you for choosing the most qualified experts in bone dosimetry as the reviewers of our manuscript. We submit a revised version of the manuscript that addresses the points raised during the review process. We have revised the title because both reviewers recommended aligning it with the scope of the manuscript. In addition, following the PLOS ONE Data policy and Reviewer #1 evaluation, we have provided a supporting information file to make the data underlying the findings described in the manuscript fully available. Below are our responses to each of the items raised by the reviewers.

Reviewer #1 (Derek W. Jokisch):

The manuscript presents an approach for computing dose factors to active marrow from intakes of Sr-89 and Sr-90 in the human femur. The authors describe construction of a synthetic, digital femoral model which is then coupled to a Monte Carlo transport code to model electron energy deposition from strontium radionuclides in the bone volume. The paper is thorough and generally well described. I recommend revisions, several of which fall into the 'major' category. I opted for "minor revisions" overall. While I have a large number of comments and suggestions, I don't believe any necessitate repeating portions of the study.

Major comments

1. As described by the authors, the paper presents a dosimetry model and methodology applied to active marrow dose from Sr-89 and Sr-90 present in the human femur. The current title is, however, broader in scope. The authors should consider revising the title to specifically include mention of the femur and the two strontium radionuclides. They could also include mention of active marrow dose, since they acknowledge in the paper that the method does not provide dose estimates to the bone endosteum, an important component of "skeletal dosimetry" in radiation protection. It is certainly appropriate for the authors to write that the same methodology presented in the manuscript can be extended to other skeletal sites and other beta emitters, as they have done. But, the current manuscript is limited to a single skeletal site and two radionuclides.

We have revised the title in order to make it more specific: “Stochastic Parametric Skeletal Dosimetry model for humans: General approach and application to active marrow exposure from bone-seeking beta-particle emitters”.

Our motivation for developing the SPSD model was the necessity for dosimetric support of Urals epidemiologic studies which revealed the effects of radiation on the hematopoietic system (as described in Lines 47-58 of the Introduction). The current limitations of the SPSD model are explicitly formulated by the authors in Lines 458-463. Bone sarcomas induced by irradiation of the endosteum were not found in the epidemiologic studies of the Urals cohorts. However, if necessary, it will be possible to extend the methodology to the bone endosteum in future.

The statement of the reviewer that the methodology presented in the current manuscript is limited to a single skeletal site (femur) is incorrect. As described in our previous publications (to which we refer in the current manuscript), the methodology is applicable to all hematopoietic skeletal sites. In particular, you can find: 

• The mathematical basis for modeling described in [21] (Zalyapin et al. 2018) available in open access http://www.mathnet.ru/php/archive.phtml?wshow=paper&jrnid=vyuru&paperid=430&option_lang=eng); 

• The software Trabecula described in [76] (Shishkina et al., Health Physics 2020, 118(1):53-9); 

• Skeletal-average Dose Factors for the adult described in [20] (Shishkina et al. 2018, available in open access https://www.rad-journal.org/paper.php?id=114). 

The current manuscript is focused on the general approach to the modeling for humans in different ages. As was stated in the abstract:“Model output for the human femur at different ages is provided as an example” (Lines 34-35). To avoid the above misunderstandings in the future, we have supplemented the current manuscript with Figure 4B visualizing synthetic models generated for segments from several skeletal sites.

2. Line 164: "In the current study both microCT imaging with adequate resolution..." For the sake of reproducibility, the authors should specify what specific resolution value was determined to be "adequate resolution". The authors could also consider providing a justification for the value they used in this study. 

The term "adequate resolution" in Line 164 has been specified (as ≤ 40 µm). Earlier (Lines 158-159), it was explained that the resolution of microCT images is evaluated as relative to real trabecular size. An example of the distribution of real trabecular sizes was shown in Fig. 2 (based on data published in [28]). According to Table 2 from [28], only 3.14% of trabeculae have thicknesses less than 50 µm. Based on this example and also other data, resolution values of ≤ 40 µm were determined to be adequate.

3. Lines 174-176: "Measured data on vertebra-cortical thickness for children were not found. However, bone images available in [49] show that cortical thickness of infant- and child- vertebra is very thin and comparable to Tb.Th." The authors should state how, or if, they applied this estimate of cortical thickness in the vertebra of children to the femur. I'm also curious how the lack of data for this important parameter folds in to the authors' uncertainty analysis. 

References [66-70] to morphometric data on femur (used in our femoral model) were given in Line 279. And they do not coincide with references [40-49] for the vertebra data shown in Table 2 (used in the vertebral model). The reviewer's assumption that the data for vertebra were used in femoral model is incorrect.

Evaluating published morphometric data for the vertebrae (Table 2), we stated that the sources of data for newborns and infants are very limited (Lines 172-176). We have expanded the discussion on the data of Table 2 to indicate what assumptions should be made for modeling and uncertainty estimation in the case of data paucity (see the revised manuscript).

4. The biokinetic model utilized (reference [15]) includes both bone volume and surface compartments. The energy deposition model used by the authors assumes uniform activity distribution throughout the bone volume. If significant Sr decays take place on the bone surface, the authors would have a problem. Presumably, the authors are okay with their approach because Sr removal rate from the bone surface compartments is fast and any resulting non-uniformity of activity throughout the bone volume is negligible. If true, the authors should consider describing this in the paper. It might also be helpful to describe the early times following intakes in children. In other words, how soon following an intake is the assumption of uniform Sr activity distribution in bone volume a valid one given the biokinetic model in [15]? 

The reviewer raises here an issue on matching dosimetric and biokinetic models which is important but beyond the scope of the current manuscript.

The reviewer correctly presumed the authors’ position that “Sr removal rate from the bone surface compartments is fast and any resulting non-uniformity of activity throughout the bone volume is negligible”. The role of bone surfaces and bone volume in Sr biokinetics was studied in detail in the second half of the 20th century. There are two processes that determine Sr uptake in bone volume: new bone formation and diffusion of ions with bone fluid through the lacuno-canaliculi system of mature bone. The cells on the bone surfaces are considered to be a fast-transit compartment, which practically does not affect Sr uptake in the bone volume. This was described in the classical publications:

ICRP, 1973. Alkaline Earth Metabolism in Adult Man. ICRP Publication 20. Pergamon Press, Oxford.

Groer PG, Marshall JH. Mechanisms of calcium exchange at bone surfaces. Calc Tiss Res 12:175-192; 1973.

Marshall J.H, Onkelinx C. Radial diffusion from canaliculi: an explanation for the retention of alkaline earths in adult bone. ANL-7360. ANL Rep: 110-114; 1966.

Our age-dependent biokinetic model (reference [15]) is based on the same assumptions and has the same compartments as R. Leggett’s biokinetic model (1992) used in ICRP-67 (1993). It should be noted that each component of compartmental models “is considered to be homogeneous, instantly mixed, with uniform concentration”. See, for instance: 

Bassingthwaighte JB, Butterworth E, Jardine B, Raymond GM. Compartmental modeling in the analysis of biological systems. Methods Mol Biol. 2012; 929:391-438. doi: 10.1007/978-1-62703-050-2-17. PMID: 23007439. 

Regarding source regions in the dosimetry model, we follow the definition given on page 39 of ICRP-110 (2008), which states: “…no distinction is made between surface and volume sources in cortical and trabecular bone. The results from the respective volume sources will be applied to estimate values for the surface sources.”

Thus, we have not suggested anything new on the issue raised (that should be discussed in the current manuscript) in comparison with the above publications.

5. Lines 235-247 and lines 313-353: Are all the trabecular rods in the synthetic model connected? And, how do they interface with the cortical shell model? The latter could be particularly important for cortical bone volume sources. If available, an image of the synthetic model analogous to Figure 2a could be helpful to readers, particularly if the image could show a region intersecting with the cortical shell. 

The stages of model generation were described in [20, 76]. In the first stage, the synthetic model of infinite spongiosa is generated in which all trabecular rods are connected. Then, a certain part of the modeled spongiosa is cut out in such a way as to fill the volume of the stylized bone segment and is voxelized. In this stage some rods can become “disconnected”. In the last stage, a cortical layer of a given thickness is formed by replacing the outer layers of the spongiosа voxels with the voxels of the cortical bone. As a result, some trabecular rods contact with the cortex and others do not in a region intersecting with the cortical shell.

A brief description of model generation stages has been added and Figure 4B has been included which presents the cross-sections of generated and voxelized models of three segments from vertebra, femur and iliac crest (on the same scale). 

6. Did the authors attempt to measure the important skeletal parameters after voxelization of their synthetic model? In other words, were the skeletal size parameters (Tb.Th, Ct.Th, BV/TV, etc.) used to create the synthetic model retained after voxelization? 

Yes, absolutely. The spongiosa model was calibrated to fit the main input parameters after voxelization. Thus, the values of Ct.Th, Tb.Th and BV/TV of the voxelized models are equal to the input model parameters (see Supporting information files that have been added to the manuscript).

7. Line 367: "The energy dependencies were compared with published results [78] based upon direct data of micro-images of bone structures. The comparison showed a similarity in the energy dependencies which was a good Quality Assurance test for our model [61]. This confirmed that the generated phantoms are dosimetrically equivalent to real bone structures in the considered range of electron energies." It would be appropriate to provide a plot showing this comparison. The conclusion re: "dosimetrically equivalent" is difficult to make in a publication without the supporting evidence. 

The plots (in English and in Russian) showing this comparison were already published in the cited references as Fig. 4 in [61] (Degteva et al. 2019) and the supporting evidence is available in open access (https://www.radhyg.ru/jour/article/view/615/617).

8. Lines 463-465: "...active marrow was considered to be the radiosensitive skeletal tissue (while bone endosteum was not considered here). Thus, it is not reasonable to compare the "specific" model suggested here with the "universal" image-based (voxel- and mesh-type) reference computational phantoms developed for ICRP [80,81]." The authors might consider expanding upon the endosteum comment. For example, why this target is not important in their application.

Comparisons between the authors work would not be made with ICPR mesh or voxel phantoms since they are not used to compute energy deposition in active marrow from bone volume sources. More appropriate would be comparisons to specific absorbed fractions for the adult published in ICRP Pub. 133 for bone volume sources irradiating active marrow. Or, a comparison with dose coefficients to the active marrow for Sr-89, 90 published in the ICRP OIR series. Given the authors concluding statements on lines 524-525 and 43-44, such comparisons seem desirable. The values above, of course, represent activity and doses in the whole skeleton, so could not be made with the femoral data presented in this work. But, the authors could compare to the femoral data in Hough et al (their ref [78]) which is the basis for the ICRP SAFs. 

Endosteum is less important in dosimetric support of epidemiological studies of the Ural cohorts, because these studies did not reveal any biological effects associated with this target irradiation.

Energy dependencies of absorbed fractions calculated using the generated vertebral model for the AM of adult were already compared to the data from Hough et al. (ref [78]). This comparison was described in peer-reviewed article [61] which is publicly available at https://www.radhyg.ru/jour/article/view/615/617. Therefore the reviewer’s recommendation was already implemented previously.

9. Lines 470-472, 529-530: Given the sparsity of data the authors found for skeletal size parameters in children, particularly young children, would the current study not also benefit from additional measurements in cadavers? 

Yes of course. If additional measurements in cadavers (as well as new in vivo studies) are released, they can be used to refine our current estimates.

Minor comments

a. In the abstract (line 38), the authors write: "The biokinetic model results will be used to calculate individual doses to members of a cohort exposed to 89,90Sr from liquid radioactive waste..." The authors should consider rephrasing to limit this to femur doses or to place after a reference to 'further' or 'future study'. As currently written, a reader would reasonably expect to find skeletal-wide doses. 

The sentence has been rephrased as: “The biokinetic model results will be used in the future to calculate…”

b. Introduction, third paragraph (lines 66-67): Please consider improving/rephrasing: "Strontium intakes are incorporated... and the Sr become a source of beta radiation for the marrow." 

The sentence has been rephrased as: “Calcium-like 89Sr and 90Sr incorporated in the mineralized bone are a source of beta radiation for the marrow”.

c. Introduction, third paragraph (line 70): "... so called "chord-based" methods..." While other papers have used the term 'chord', Darley pointed out the more appropriate term is 'path length' or 'path-length'. See Darley, Philip J. Health Physics. 90(2):176-177, February 2006 and the response to Darley by Shah, Amish P.; Rajon, Didier A.; Jokisch, Derek W.; Health Physics. 90(2):177-179, February 2006. 

The term ‘chord-based’ has been changed to 'path length-based'.

d. Introduction, third paragraph (line 71): Reference [18] does not describe radiation transport in a voxelized skeletal model. Reference [78] (Hough et al.) does and the authors might consider citing it, here. The authors should also consider citing voxel studies published by Kramer et al. and Gao et al.

ELECTRON ABSORBED FRACTIONS IN AN IMAGE-BASED MICROSCOPIC SKELETAL DOSIMETRY MODEL OF CHINESE ADULT MALE

Shenshen Gao, Li Ren, Rui Qiu, Zhen Wu, Chunyan Li, Junli Li

Radiation Protection Dosimetry, Volume 175, Issue 4, August 2017, Pages 450–459, https://doi.org/10.1093/rpd/ncw372

Skeletal dosimetry based on µCT images of trabecular bone: update and comparisons

R Kramer, V F Cassola, J W Vieira, H J Khoury, C A B de Oliveira Lira and K Robson Brown Physics in Medicine & Biology, Volume 57, Number 12 Citation R Kramer et al 2012 Phys. Med. Biol. 57 3995

Reference [18] (Jokisch et al. 2001) has been changed to (Kramer et al. 2012)

e. Lines 83-84: "In the current study, the spongiosa contains both source areas (bone trabeculae) and target areas (active and inactive marrow)." I believe the authors intend for the source regions to be both cortical and trabecular bone. Similarly, inactive marrow is not the target of interest, though the authors later explain that they have approximated the dose to the entire marrow space (active and inactive) as being equivalent to the active marrow dose. 

The sentence has been rephrased as: "The spongiosa contains both source areas (bone trabeculae) and target area (AM). The cortical shell is an additional exposure source for the AM. ".

f. Line 130: "Different methods were used to evaluate the parameters, each of which has its pros and cons and is not completely accurate due to the complexity of the 3D bone structure." The authors should consider improving this sentence. It is not clear what basis was used to "evaluate the parameters" or what quantity is being referenced as "not completely accurate". The sentence is too generalized as currently written. 

The sentence has been rephrased as: "Different methods were used to evaluate the parameters, each of which has its pros and cons [26,27]. A brief evaluation of the methods is given below."

g. Line 161: "Chappard et al. note that ... depend on the three-dimensional reconstruction method used in histomorphometry (better agreement is observed for the rod trabecular models)." The authors may wish to revisit this sentence. I found the parenthetical confusing... better agreement to what? Perhaps the parenthetical is out of context and can be deleted? The authors may also wish to revisit the next sentence, ("At the same time...") as well. 

The parenthetical has been deleted. The next sentence has also been edited.

h. Line 211: "The marrow-filled cavities... are considered to be the target regions." While the authors later describe that dose to the bone endosteum is not computed in this model, the authors should also consider pointing it out here, at first use. 

The revised title now specifies that only active marrow (but not endosteum) is considered in the current manuscript. There is no need to repeat this in the text.

i. Line 213: "...the density of AM and IM is almost the same..." The two densities are roughly 5% different. The authors should consider replacing 'almost the same' with something more specific about the difference, as they have been elsewhere in the paper when similar sized differences exist. 

Values of the two densities were explicitly specified earlier in Line 194. Thus the expression in Line 213 has been deleted as redundant.

j. Line 247: The authors should consider replacing "dosimetric equivalence" with a different phrase or description which does not require quotation marks. 

The sentence with the term "dosimetric equivalence" has been deleted because it was not entirely appropriate for this section of the manuscript. The issue is discussed in detail below in the section ‘Stochastic generation of bone segment phantoms’.

k. Line 261: "Sex-differences ... were identified for people over 15 years old." Does this mean sex differences only exist in the adult models? If so, the authors should consider rephrasing the sentence or adding that conclusion. 

The sentence has been rephrased as: “Sex-differences in skeletal macro-parameters were considered for people starting in adolescence [52].”

l. Line 320: The authors should consider improving the sentence which begins: "The rod-like trabeculae along the edges..." 

Sentences at lines 316-22 revised as follows:

The spongiosa microarchitecture is simulated with interconnected rod-like structures representing trabeculae. Figure 4A illustrates a model example of four interconnected trabeculae. The spongiosa model results from deformation of a three-dimensional grid by stochastic node perturbation and random variation of the node thickness. The rod-like trabeculae, now simulated with cone segments resulting from varying end-node thicknesses, are either truncated along the edges of the deformed grid or connected with other trabeculae within the deformed grid using spheres [20,21].

m. Lines 385-388: The authors should consider rewriting these two sentences. Specifically, perhaps the order should be switched? e.g. The method is ... The code "Trabeculae" implements that method. 

Sentences at lines 385-88 (after the sentence “It should be noted that, for each age, the AM fraction in the total marrow within the hematopoietic site is considered the same for all segments”) revised as follows:

Use of this approach results in weighting factors that are proportional to segment-specific marrow volume. Segment-specific bone marrow volumes are calculated by the code “Trabecula”.

n. Line 389: "The calculated marrow volumes showed that..." The authors should consider rewriting this sentence to state that the model is consistent with AM redistribution throughout life. The model itself is not evidence of that redistribution. 

Sentence at lines 389-90 revised as follows:

The calculated marrow volumes are consistent with the knowledge that throughout human life a significant redistribution of AM occurs between different segments inside the femoral site (see also Table 1).

o. Line 412: "...where the crossfire dose rate is..." The authors should consider defining what is meant by "crossfire dose rate".

The word “crossfire” has been deleted.

p. Line 414: Add units to the dose factor values at the end of the sentence. 

The units have been specified above in Line 402.

q. The authors should look out for multiple places in the text on and on figures where the dose to the marrow is specific to the femur only. Please consider adding the word "femur" in such locations so the reader does not get confused. The right axis on Figure 6 and the left axis on Figure 5-bottom seem particularly important places to add 'femur' or 'femoral'. 

We have checked the places in the text and on figures where the dose to the marrow is specific to the femur only. Section title ‘Preliminary modeling results: some examples’ has been changed to ‘Preliminary modeling results: Femur as an example’. Term ‘femoral AM’ is explicitly present in the legends to Figures 5 and 6. In our opinion, after the final formatting (when the legends will go directly after Figures 5 and 6), the word 'femoral’ in the axis names will already be redundant.

r. Line 506: Would "largest intake" be more appropriate than "maximum intake"?

Yes, we have changed "maximum intake" to "largest intake"

s. Lines 520-522: Please consider rephrasing or removing the sentence beginning with "Thus, for the analysis of the excess..." This falls into the future work category, correct? 

The sentence has been rephrased as: "Thus, for the future analysis of the excess..."

t. Line 214: "The cortical bone volume (CBV) is considered as a source of crossfire exposure of the marrow. Thus, the dosimetry problem is to evaluate energy deposition in target regions for particles emitted in TBV and CBV sources." Should the first sentence also include TBV? Should target "regions" be "region" since the authors consider one target (the marrow)?

Yes, the sentences have been changed as recommended.

Reviewer #2 (Wesley E. Bolch):

General Comments

This submitted paper describes a new and innovative approach to skeletal dosimetry with specific application to bone-seeking beta-particle emitters. The application is general to all beta-emitter bone dosimetry assessments but the specific purpose for this model development is The approach – while initially simplistic in its description of bone morphometry – is upon further reflection quite elegant and appropriate to the application of active marrow dose assessment. The vast majority of skeletal dosimetry models over the decades have been based upon direct imaging of cadaveric skeletal samples – from 2D optical scanning of bone slice radiograph to 3D imaging by NMR microscopy and microCT scanning. The authors of this report have taken a different approach – to model both the skeletal macrostructure and microstructure via stylized computational approaches – simple geometric shapes. They argue – quite successfully – that the simplicity of the morphometric models does not detract from dose accuracy, but at the same time, these types of geometric models allow for a stochastic sampling of a number of dosimetrically-important features to include tissue macrostructural dimensions (such as the thickness of the bone site cortex of cortical bone), tissue mixtures (marrow cavities and bone trabeculae within spongiosa), elemental compositions, tissue compositions (such as the mixture of hematopoietically active versus inactive bone marrow), and tissue microstructural dimensions (such as the thicknesses of the bone trabeculae and marrow cavity spaces). With this ability to “change” the model, following detailed collection of literature values and their statistical uncertainties and ranges, a fully stochastic bone dosimetry model can be implemented. For a purely image-based modeling approach this would require a significant cost and time for multiple cadaveric tissue collections and imaging sessions. One limitation of the model is its application to the pediatric child due to a paucity of data on bone morphometry and tissue features – but this is clearly not the fault of the modelers, and they do the best they can with this limited data source.

The paper is very well-written, clearly organized, and highly referenced. I have no specific editorial changes to make.

Thank you very much!

I would like the authors however to comment on the following issues:

• The title of the article might include the phrase – “General descriptions and applications to the femur” as most of the dosimetry data presented here is limited to this one bone site. 

We have revised the title in order to make it more specific: “Stochastic Parametric Skeletal Dosimetry model for humans: General approach and application to active marrow exposure from bone-seeking beta-particle emitters”. The current manuscript is focused on a general methodology which is applicable to all hematopoietic skeletal sites. This also was described in our previous publications [20, 21]. Computational results for the femur are provided in the current manuscript as an example of the methodology application. 

See also our response to Reviewer#1 major comment 1.

• The authors should further justify why voxelized models are used in the final radiation transport simulations. Could not they present their model in a polygon mesh format with direct coupling to the transport code? This would fully eliminate any voxel-size effects in the dosimetric results. 

Transformation of a mathematical model of bone microstructure into good quality mesh seems to us a more complex procedure than just voxelization (with resolution smaller than Tb.Th). Actually, the limitation of voxel models in terms of realism is not important for the synthetic microstructure fitted into the stylized shape. Therefore, at the current stage of the study we decided to simplify the task making sure that the input parameters match the output parameters after voxelization. However, we agree that the further direction of model improvement may be the increase of realism of bone shape and cortex description with the use of a polygon mesh format.

• What geometric expressions are to be used for more complex bone sites such as the cranium, mandible, vertebrae, and pelvis? 

Complex bones were divided into segments described by simple geometric shapes. For example, the flat bones of the skull with a large surface area were truncated to a box with a base of 3x3 cm. The bones of the vertebrae and pelvis were divided into many small segments (mainly with rectangular and cylindrical shape). 80% of all segments were boxes and regular cylinders. The rest are the deformed cylinders and triangular prisms. Segmentation was described in more detail in [62] (Sharagin et al. 2018 DOI: 10.21175 / RadProc. 2018. 33; open access). In the near future, we plan to publish phantoms of people of different ages and genders with a full description of all segments.

• For high-energy beta emitters in bone, there will be bremsstrahlung x-ray production and cross-bone site irradiation. How will this be handled within the SPSD model? 

The effect of cross-bone site irradiation was estimated by the comparison of the model prediction with calculations based on the model with extended linear dimensions (by 2 mean path lengths of electrons (in spongiosa) in the directions of the contact with other sites). This topic was pointed out in [79] (Shishkina et al. 2019). Currently we are just finalizing this study. It is found that some segment-specific DFs should be corrected by up to a factor of 1.24. However, the overall effect of cross-bone site irradiation on the skeleton-average DF(AM←TBV) is about 2-5% Therefore, this topic is not included in this paper on the basic methodology (to escape the overload). The details of the study will be published separately.

• Can the authors make a brief comment on final dosimetry results feed to the radiation epidemiologists? What will they do with stochastically sampled active marrow bone doses? Still use only the central estimates of bone marrow dose or will the dosimetry uncertainties carry through to cancer risk uncertainties? 

The modeling results will be incorporated into a special Monte Carlo dosimetry system that provides multiple realizations of individual dose estimates for each member of an epidemiological cohort (Napier et al. 2013). Thus, confidence intervals in cancer risk estimates for the cohort can be corrected by the radiation epidemiologists in accordance with the dosimetry uncertainties (Zhang et al. 2017). 

Napier B.A., Degteva M.O., Shagina N.B., Anspaugh L.R. (2013) Uncertainty analysis for the Techa River Dosimetry System. Medical Radiology and Radiation Safety 58(1):5–28. 

Zhang Z, Preston DL, Sokolnikov M, Napier BA, Degteva M, Moroz B, et al. (2017) Correction of confidence intervals in excess relative risk models using Monte Carlo dosimetry systems with shared errors. PLoS ONE 12(4): e0174641. https://doi.org/10.1371/journal.pone.0174641

Specific Comments

None – very well written and informative paper. My congratulations to the authors.

Thank you very much!

Sincerely,

Bruce A. Napier

---

## [Decision Letter · Decision Letter 1]

6 Sep 2021

Stochastic Parametric Skeletal Dosimetry model for humans: General approach and application to active marrow exposure from bone-seeking beta-particle emitters

PONE-D-21-09605R1

Dear Dr. Napier,

We’re pleased to inform you that your manuscript has been judged scientifically suitable for publication and will be formally accepted for publication once it meets all outstanding technical requirements.

Kind regards,

Derek Jokisch

Guest Editor

PLOS ONE

Additional Editor Comments (optional):

In order to preserve transparency, please note that I served as a reviewer on the first submission of your paper.  Upon receipt of your revision, the editorial staff at PLOS ONE asked me to serve as Guest Academic Editor for your paper.

Reviewers' comments:

Reviewer's Responses to Questions

**Comments to the Author**

1. If the authors have adequately addressed your comments raised in a previous round of review and you feel that this manuscript is now acceptable for publication, you may indicate that here to bypass the “Comments to the Author” section, enter your conflict of interest statement in the “Confidential to Editor” section, and submit your "Accept" recommendation.

Reviewer #2: All comments have been addressed

2. Is the manuscript technically sound, and do the data support the conclusions?

Reviewer #2: Yes

3. Has the statistical analysis been performed appropriately and rigorously? 

Reviewer #2: Yes

4. Have the authors made all data underlying the findings in their manuscript fully available?

Reviewer #2: Yes

5. Is the manuscript presented in an intelligible fashion and written in standard English?

Reviewer #2: Yes

6. Review Comments to the Author

Reviewer #2: The authors have satisfactorily address all my prior comments and questions. The revised paper is thus ready for final editorial decision on publication.

7. PLOS authors have the option to publish the peer review history of their article (what does this mean?). If published, this will include your full peer review and any attached files.

Reviewer #2: No

---

## [Editor Report · Acceptance letter]

7 Oct 2021

PONE-D-21-09605R1 

Stochastic Parametric Skeletal Dosimetry model for humans: General approach and application to active marrow exposure from bone-seeking beta-particle emitters 

Dear Dr. Napier:

I'm pleased to inform you that your manuscript has been deemed suitable for publication in PLOS ONE. Congratulations! Your manuscript is now with our production department. 

Kind regards, 

on behalf of

Dr. Derek Jokisch 

Guest Editor

PLOS ONE